



# A Model Intercomparison of Stratospheric Solar Geoengineering by Accumulation-Mode Sulfate Aerosols

Debra K. Weisenstein[1,5], Daniele Visioni[3], Henning Franke[4], Ulrike Niemeier[4], Sandro Vattioni[2], Gabriel Chiodo[2], Thomas Peter[2], David W. Keith[1]

5  [1]Harvard John A. Paulson School of Engineering and Applied Sciences, Cambridge, MA-02138, USA
[2]Institute of Atmospheric and Climate Science, ETH Zürich, Zurich, 8002, Switzerland
[3]Sibley School of Mechanical and Aerospace Engineering, Cornell University, Ithaca, NY-14850, USA
[4]Max Planck Institute for Meteorology, Hamburg, 20146, Germany
[5]Retired

10  *Correspondence to*: David Keith (david_keith@harvard.edu)

**Abstract.** Analyses of stratospheric solar geoengineering have focused on sulfate aerosol, and almost all climate model experiments on sulfate aerosol have assumed injection of $SO_2$. Yet continuous injection of $SO_2$ may produce overly large aerosols. Injection of $SO_3$ or $H_2SO_4$ from an aircraft in stratospheric flight is expected to produce new accumulation-mode particles (AM-$H_2SO_4$), and such injection may allow the sulfate aerosol size distribution to be nudged towards higher radiative efficacy. We report the first multi-model intercomparison of AM-$H_2SO_4$ injection. We compare three models: CESM2(WACCM), MAECHAM5-HAM, and SOCOL-AER coordinated as a testbed experiment within the Geoengineering Model Intercomparison Project (GeoMIP). The intercomparison explores how the injection of new accumulation-mode particles changes the large-scale particle size distribution and thus the overall radiative and dynamical response to sulfate aerosol injection. Each model used the same injection scenarios testing AM-$H_2SO_4$ and $SO_2$ injections at 5 and 25 Tg(S) yr$^{-1}$ to test linearity and climate response sensitivity. All three models find that AM-$H_2SO_4$ injection increases the radiative efficacy, defined as the radiative forcing per unit of sulfur injection, relative to $SO_2$ injection. Increased radiative efficacy means that when compared to the use of $SO_2$ to produce the same radiative forcing, AM-$H_2SO_4$ emissions could reduce some side-effects of sulfate aerosol geoengineering such as stratospheric heating. We explore the sensitivity to injection pattern by comparing injection at two points at 30° N and 30° S to injection in a belt along the equator between 30° S and 30° N, and find opposite impacts on radiative efficacy for AM-$H_2SO_4$ and $SO_2$, suggesting that prior model results for concentrated injection of $SO_2$ may be strongly dependent on model resolution. Model differences arise from differences in aerosol formulation and differences in model transport and resolution, factors whose interplay cannot be easily untangled by this intercomparison.



# 1 Introduction

Deliberate modification of Earth's albedo has been proposed to counteract some of the radiative forcing from the rise in $CO_2$ and other greenhouse gases (GHG) caused by human emissions (Budyko 1974; Crutzen 2006). Despite the complexity of the climate system and certainty that its manipulation carries risks, solar radiation modification (SRM) is being studied with

climate models to examine the potential benefits and risks while simultaneously improving our knowledge of climate interactions and feedback processes. The most studied SRM proposals involve a deliberate enhancement of the Earth's stratospheric sulfuric acid aerosol layer by injection of sulfuric acid aerosol (also called sulfate aerosol) or its precursor gases into the stratosphere. Potential SRM scenarios could be effective by slowing the rate of change of climate over decades to a century, allowing time for emission mitigation, adaptation, or GHG removal.

Many SRM model experiments have used alteration of the solar constant as a simple proxy for exploring the climate response to stratospheric aerosol injection. Of the GCM studies that have explicitly simulated alteration of stratospheric aerosols, almost all have either injected $SO_2$ or directly prescribed an increase in the sulfate aerosol burden. Simulations of $SO_2$ injection are motivated, in part, from an analogy to volcanoes, which are found to alter climate as a result of the increase in stratosphere aerosol loading (Robock, 2000). Volcanic injections and their effects on climate have been a major

motivation for the inclusion of stratospheric sulfate aerosols in global climate models. Yet, stratospheric solar geoengineering (SSG) scenarios differ from volcanic aerosol injections in that with SSG sulfur will presumably be injected into aircraft plumes, producing high local concentrations and strong gradients, and because emissions will be continuous in time, factors which will yield different microphysical behaviour (Heckendorn et al., 2009).

Studies of SSG by injection of gas phase $SO_2$ have found limitations including: (1) reduced efficacy at higher loading

due to unfavorable aerosol size distributions with possible limitations on achievable radiative forcing (Heckendorn et al., 2009; English et al., 2012; Niemeier and Timmreck, 2015), (2) depletion of stratospheric ozone (Tilmes et al. 2009; Pitari et al. 2014), (3) stratospheric heating which also perturbs stratospheric circulation and water vapor (Ferraro et al., 2011; Aquila et al., 2014; Richter et al., 2017; Niemeier amd Schmidt, 2017; Franke et al., 2021), (4) enhanced diffuse light at the Earth' surface (Kravitz et al., 2012), and (5) impacts on upper tropospheric ice clouds (Kuebbeler et al., 2012; Visioni et al., 2018a).

These limitations might be addressed through use of various solid aerosol particles for SSG (e.g. Pope et al., 2012; Weisenstein et al., 2015; Keith et al., 2016); alternatively some of the limits may be addressed by altering the size distribution of sulfate aerosol.

Efficacy is decreased with increased $SO_2$ injection because most of the added sulfur increases the mass of existing particles rather than forming new accumulation mode particles. Pierce et al. (2010) proposed that addition of accumulation-

mode particles could be used to steer the overall large-scale aerosol size distribution towards the size range that produces the most radiative forcing per unit mass of injected sulfur ("radiative efficacy"). The accumulation mode particles would be formed when $H_2SO_4$ or $SO_3$ vapor are released into an aircraft wake where nucleation and coagulation in the confined plume results in a distribution of sulfate particles in the accumulation size range (0.05-0.2 μm radius). The impact of this proposed



methodology was subsequently tested in a global three-dimensional model of aerosol microphysics by English et al. (2012), who found larger stratospheric sulfate burdens with injection of accumulation-mode particles rather than equivalent emissions of $SO_2$ or gas phase $H_2SO_4$. Later, Vattioni et al. (2019) used a three-dimensional interactive chemistry-climate-aerosol model and found improved radiative efficacy of SSG by accumulation-mode sulfate aerosol injection over that of

$SO_2$ injection.

The Geoengineering Model Intercomparison Project (GeoMIP) was formed in 2011 to coordinate a common set of experiments for the purpose of assessing climate model responses and sensitivities to solar radiation management (Kravitz et al., 2011; Kravitz et al., 2015). GeoMIP scenarios have included uniform reductions in solar radiation as well as specified injections of $SO_2$ into the tropical stratosphere or specified aerosol distributions (Tilmes et al., 2015). This study represents a

GeoMIP testbed experiment in which three of the participating GeoMIP models ran identical scenarios exploring the impacts of controlled accumulation-mode sulfate aerosol injection, which we refer to as AM-$H_2SO_4$ injection, into GCMs.

The evolution of aerosol particles after injection of $H_2SO_4$ includes the initial formation of nucleation mode particles by homogeneous nucleation of $H_2SO_4$ gas and the subsequent formation of accumulation-mode particles by coagulation of the nucleation mode. Our study does not address these plume-scale microphysical processes. We simply specify a size

distribution of $H_2SO_4$ aerosol that is delivered at the scale of each model's numerical grid. Our aerosol size distribution is consistent with Pierce et al. (2010) and Benduhn et al. (2016) who modelled plume microphysics and found that injection rate could be adjusted to produce sulfate aerosol size distributions in the 0.1-0.15 µm radius size range. For the temporal and spatial scale beyond plume models, global GCMs such as the GeoMIP models can be used to simulate injections of the given size distribution of sulfate aerosols into their grid cells. These GCMs can effectively simulate changes in global aerosol

burden, radiative forcing, ozone, and stratospheric temperature and circulation. As input, they take the particle size distributions from aircraft plume model studies.

Three GCMs with interactive aerosol microphysics participated in this experiment: the National Center for Atmospheric Research (NCAR) Community Earth System Model (CESM2) with the Whole Atmosphere Community Climate Model (WACCM) atmospheric configuration, the Max-Planck Institute's middle atmosphere version of ECHAM5 with the HAM

microphysical module (MAECHAM5-HAM), and SOCOL-AER version 2 model. The CESM2 and MAECHAM5-HAM models employ a modal scheme to prescribe the aerosol size distributions, while the SOCOL-AER model uses a sectional scheme. The CESM2 and SOCOL-AER models interactively couple the aerosol and ozone through photochemistry and heterogeneous reactions whereas the MAECHAM5-HAM model uses prescribed and precalculated ozone and OH concentrations when calculating sulfur chemistry to predict aerosol concentrations. The MAECHAM5-HAM and CESM2

models internally generate a quasi-biennial oscillation (QBO) while the SOCOL-AER model uses nudging to simulate QBOs. As this study focuses on stratospheric responses to the aerosol injections, all were run with specified sea surface temperatures, simplifying the interpretation of inter-model differences. Section 2 includes a description of these models and of the scenario calculations.



## 2 Description of models and emission scenarios

All three models in this inter-comparison are three-dimensional dynamic general circulation models. All include sulfur chemistry and interactive microphysics for prognostically calculating the size distribution of stratospheric aerosol. All
models allow radiative heating from aerosols to alter dynamics and thus the transport of aerosols and trace gases. Table 1 summarizes the most relevant aspects of the models. The CESM2 (Donabasoglu et al., 2016) and MAECHAM5-HAM (Stier et al., 2005) models employ a modal representation of the aerosol size distributions, utilizing 3 (for CESM2) or 4 (for MAECHAM5-HAM) lognormal modes to describe the size range of sulfate aerosols from nanometer to micron scale. CESM2 includes 4 modes total but only 3 modes represent sulfate aerosol (Liu et al., 2016). Their schemes differ in the size
ranges and assumed lognormal width, σ, of the modes, with the CESM2 model utilizing a coarse mode with radius greater than 0.5 μm whereas the MAECHAM5-HAM model considers the coarse mode with radius greater than 0.2 μm. The SOCOL-AER version 2 model (Feinberg et al., 2019; Sheng et al., 2015) employs a sectional aerosol scheme with 40 bins representing dry radii from 0.39 nm to 3.2 μm. Either scheme has been shown capable of prognostically generating realistic aerosol distributions (Weisenstein et al., 2007; Kokkola et al., 2009), though modal schemes require a priori assumptions on
the width of the lognormal modes which may differ for background and perturbed conditions. Sectional schemes suffer from numerical diffusion in size space.





**Table 1: Models used in this study, their horizontal resolution, number of levels and model top height, aerosol formulation, dynamical core, chemical interactivity and QBO interactivity.**

| Model | Horizontal Resolution | Vertical Levels | Sulfate Aerosol Formulation | Chemistry and Dynamics | QBO |
|---|---|---|---|---|---|
| CESM2 (WACCM) | 0.95° x 1.25° | 70 levels to $6 \times 10^{-6}$ hPa | 3 modes (Aitken, Accumulation, Coarse) | CAM dynamical core  Interactive chemistry and $O_3$ | Interactive |
| MAECHAM5-HAM | T42 (2.8° X 2.8·) | 90 levels to 0.01 hPa | 4 modes (nucleation, Aitken, Accumulation, Coarse) | ECHAM5 dynamical core  Fixed OH and $O_3$ | Interactive |
| SOCOL-AER | T42 (2.8° X 2.8·) | 39 levels to 0.01 hPa | 40 sections (0.4 nm to 3.2 µm dry radius) | ECHAM5 dynamical core  Interactive chemistry and $O_3$ | Nudged |

5  The MAECHAM5-HAM (Niemeier et al., 2020; Niemeier and Timmreck, 2015; Stier et al., 2005) and SOCOL-AER models (Feinberg et al., 2019; Stenke et al, 2013) share the same dynamical core from MAECHAM5 (Roeckner et al., 2003; Roeckner et al., 2006) and used the same horizontal resolution (T42 or 2.8°x2.8° in longitudes and latitudes) and model top (0.01 hPa or approximately 80 km). However, the MAECHAM5-HAM model uses 90 vertical levels and internally generates a quasi-biennial oscillation (QBO), which has been found in several studies to modify the effects of

10  geoengineering injections (Aquila et al., 2014; Richter et al., 2017; Niemeier et al., 2020, Franke et al., 2021). The SOCOL-AER model uses only 39 vertical levels and employs nudging to reproduce a QBO that does not vary with geoengineering scenario. The CESM2 model (Donabasoglu et al., 2016) in the WACCM6 configuration (Gettelman et al., 2019) uses a finer horizontal resolution (0.95°x1.25° in longitude and latitude) than the other models and has a higher model top (6 x $10^{-6}$ hPa or approximately 130 km) with 70 vertical levels. The vertical resolution of CESM2 allows for an internally-generated QBO.



SOCOL-AER employs fully interactive chemistry from MEZON (Stenke et al., 2013), while the MAECHAM5-HAM model includes $SO_2$ oxidation chemistry only with OH, $NO_2$, and $O_3$ concentrations prescribed, thus missing potential chemical-dynamical feedbacks due to geoengineering injections. The version of CESM2 used here has a reduced set of tropospheric reactions, but full interactive chemistry in the stratosphere, mesosphere and lower thermosphere (known as the

middle atmosphere version) with 98 chemical species simulated and with prognostic stratospheric aerosols. The modal scheme here, unlike pervious versions of the same scheme, allows for growth of sulfate aerosols also in the larger size mode for a more proper representation of stratospheric processes (Mills et al., 2016).

Boundary conditions for GHGs and ozone depleting substances (ODS) use the 2040 projection values from the SSP5-8.5 scenario (O'Neill et al., 2016). All the models used a configuration with annually repeating monthly mean climatological

sea surface temperatures (SSTs) and sea ice boundary conditions derived from an average of the years 1988-2007 of the CMIP5 PCMDI-AMIP-1.1.0 SST/Sea Ice dataset (Taylor et al., 2016).

We used prescribed SSTs because not all models used in this intercomparison can utilize a coupled ocean and because prescribing SSTs allows shorter integration times to achieve the a given signal-to-noise ratio. This also simplifies interpretation of changes in radiative forcing and stratospheric temperature, since it removes differences due to the model's

climate sensitivity. The combination of SSTs averaged from 1988 to 2007 and GHGs from 2040 was selected so that, with radiative forcing from increased GHG roughly cancelled by the sulfate aerosol burden of a 5 Tg(S) yr$^{-1}$ injection, the overall model disequilibrium between atmosphere and sea surface was minimal. Models were run for ten years for each scenario, with the first two years considered spin-up and the final eight years averaged and used in our analysis.

The calculations performed for this intercomparison include a baseline or reference scenario without geoengineering and

8-12 perturbation scenarios including stratospheric aerosol injection. The perturbation scenario parameters chosen are shown in Table 2. Sulfur was injected in one of two forms, either as $SO_2$ gas or as accumulation-mode $H_2SO_4$ (AM-$H_2SO_4$) aerosol particles of specified size. The sectional model SOCOL-AER assumed a lognormal distribution for the injected AM-$H_2SO_4$ with dry mode radius of 0.1 μm, wet mode radius 0.12 μm, and mode width σ of 1.5. The modal models (CESM2 and MAECHAM5-HAM) input the AM-$H_2SO_4$ particles into their accumulation-mode: for the CESM2 model, the input size

distribution has a dry mode radius of 0.1 μm and wet mode radius of 0.12 μm with σ=1.5, while MAECHAM5-HAM has an input dry mode radius of 0.075 μm and wet mode radius of 0.1 μm with σ=1.59. Two different injection patterns were chosen: either broadly distributed between 30° S and 30° N, 19-21 km (CESM2 and SOCOL-AER) or 18-20 km (MAECHAM5-HAM), and across all longitudes (hereinafter called regional injections), or narrowly injected at two model grid points located at 30° S and 30° N, at 20 km (CESM2 and SOCOL-AER) or 18-20 km (MAECHAM5-HAM), and at

180° E longitude (hereinafter called 2point injections). All models ran with injections of 5 and 25 Tg sulfur per year (Tg(S) yr$^{-1}$), and the MAECHAM5-HAM and SOCOL-AER also ran with a 10 Tg(S) yr$^{-1}$ injection. The combination of two injection forms, two injection patterns, and two (or three) injection rates yielded 8 (or 12) perturbation scenarios. The same





set of model calculations (excluding SOCOL-AER due to lack of an internal QBO) has been analysed for changes in the QBO by Franke et al. (2021).

5 **Table 2: Each model ran a total of 8 (or 12) geoengineering scenarios, plus a reference scenario with no geoengineering. The scenarios included two injection forms (SO$_2$ gas or AM-H$_2$SO$_4$ particulate) and two or three injection rates at each of two injection locations.**

| | | |
|---|---|---|
| **Injection Form** | SO$_2$ gas | |
| | AM-H$_2$SO$_4$ particulate | • Modal models – input into accumulation mode<br>• Sectional models – lognormal distribution with dry R$_g$=0.1 μm, σ=1.5 |
| **Injection Pattern** | 2Point: 30° S and 30° N, 20 km, 180° E | |
| | Region: 30° S-30° N, 19-21 km, all longitudes | |
| **Injection Rate** | 5 Tg(S) yr$^{-1}$ | |
| | 10 Tg(S) yr$^{-1}$ (optional) | |
| | 25 Tg(S) yr$^{-1}$ | |

**3 Results**

10 We analyse changes in global aerosol properties and radiative forcing to determine whether the use of AM-H$_2$SO$_4$ can increase (compared SO$_2$) the radiative efficacy per unit of material injected across a range of models. If so, can this be attributed to increased stratospheric lifetime of the aerosols, improved scattering efficacy, or some other factor? What contributes to inter-model differences, and what can these differences tell us about uncertainty in the response to the aerosol injections? Finally, we examine some of the side effects of increasing stratospheric aerosol and explore how they differ with 15 AM-H$_2$SO$_4$ versus SO$_2$ injection and with injection pattern.

**3.1 Changes in global aerosol properties**

We start by examining the aerosol burden using the global (troposphere + stratosphere) rather than stratospheric burden to reduce uncertainties that would arise from inconsistent diagnoses of tropopause height and because the troposphere




represents less than 10% of the total burden increase in our scenarios. Injections of 5 Tg(S) yr$^{-1}$ in the form of SO$_2$ yield increases in the global aerosol burden of 2.7 to 6.6 Tg of sulfur while injections of 5 Tg(S) yr$^{-1}$ as AM-H$_2$SO$_4$ yield increases in aerosol burden of 4.2 to 8.1 Tg of sulfur (Fig. 1). Accumulation mode particle injection produces a larger burden increase than SO$_2$ injection in all cases except for the CESM2 model using 2point injections of 5 Tg(S) yr$^{-1}$. Inter-model differences

are roughly a factor of two larger than difference between SO$_2$ and AM-H$_2$SO$_4$. Differences arise in part from dynamics including the QBO which may influence microphysics by changing tropical confinement (Visioni et al., 2018b). The CESM2 model in all cases shows the highest burden increases and the MAECHAM5-HAM model the lowest in most cases. The SOCOL-AER model, with the same dynamical core as the MAECHAM5-HAM model, produces burden increases closer to that model than to CESM2. For 25 Tg(S) yr$^{-1}$ injections, most results scale proportionately, though in this case the

MAECHAM5-HAM model produces a larger burden than SOCOL-AER with AM-H$_2$SO$_4$ injections. A previous intercomparison of geoengineering results between the MAECHAM5-HAM and CESM2 models (Niemeier et al., 2020) found significantly larger aerosol burden increases for equatorial injection of SO$_2$ with CESM2 than with MAECHAM5-HAM and attributed the greater CESM2 burden to greater vertical advection in the CESM2 model. Differences in global aerosol burden due to injection pattern (2point or region) are modest except with the CESM2 model injecting AM-H$_2$SO$_4$. In

most cases, regional injections produce slightly greater global burdens.

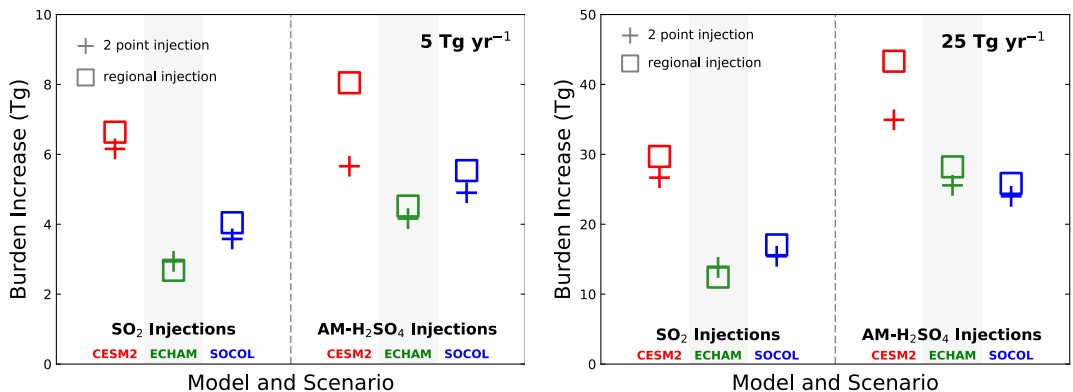

**Figure 1: Global sulfate aerosol burden increase (90° S-90° N, stratosphere+troposphere) in Tg of sulfur due to geoengineering injection of (left panel) 5 Tg(S) yr$^{-1}$ and (right panel) 25 Tg(S) yr$^{-1}$. Each panel shows the three models, CESM2 in red, MAECHAM5-HAM (labelled ECHAM) in green, and SOCOL-AER in blue, with the left side of each plot representing injection**
**as SO$_2$ and the right side of each plot representing injection as accumulation-mode H$_2$SO$_4$. Square symbols represent injection into a belt around the equator from 30° S to 30° N, 19-21 km, and all longitudes. Plus symbols represent injection into two model grid points at 30° S and 30° N, 20 km, and 180° E longitude.**

Figure 2 shows the zonal mean of vertically integrated aerosol mass increases (troposphere and stratosphere) as a function of latitude relative to the reference scenario for each model. 2Point injections at 30° S and 30° N show maximum





aerosol column burdens at about 45-50° S and 45-50° N, with sharper peaks in the southern hemisphere due to the stronger polar vortex there. Regional injections which evenly spread the injection mass in a belt around the equator between 30°S and 30°N result in aerosol column burden peaks over the equator and at 45° N and 45° S, and minima at 30° S and 30° N at the subtropical barrier zone (see e.g. Strahan and Douglas, 2004). Of these two injection patterns, the regional injections (30° S-

5    30° N) yield more uniform global distributions of aerosol, whereas the 2point injections at 30° S and 30° N concentrate more aerosol at mid and high latitudes which could concentrate geoengineering effects toward the high latitudes which are warming fastest. This figure explains the significant differences in total burden between the 2point and regional injections of AM-$H_2SO_4$ seen in the CESM2 model in Fig. 1 as a strong subtropical barrier minimizing the impact of extra-tropical injections on the tropics and tropical upwelling enhancing the impact of tropical injections in the same region. Because $SO_2$

10   injections yield larger particle sizes, the tropical upwelling has less impact on the aerosol burden in that case.

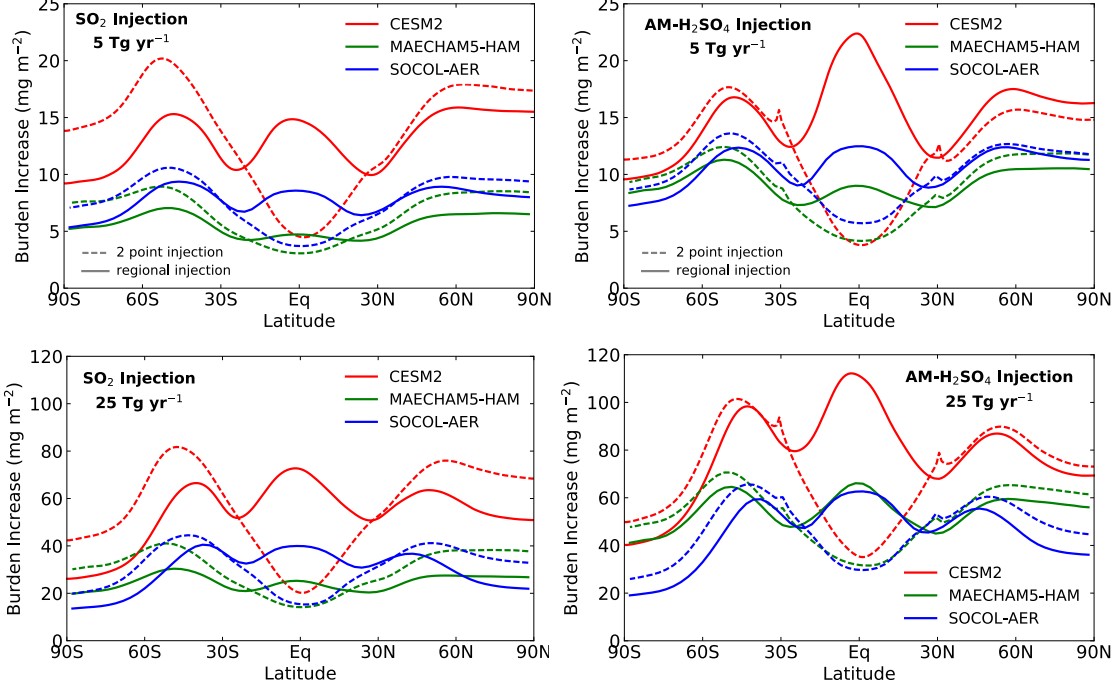

**Figure 2:** Zonal mean aerosol column burden increase above background (mg m$^{-2}$) with 5 Tg(S) yr$^{-1}$ injections (top panels) and 25 Tg(S) yr$^{-1}$ injections (bottom panels) as a function of latitude for $SO_2$ injections (left panels) and AM-$H_2SO_4$ injections (right panels). Regional injections are shown with solid lines and 2point injections with dashed lines.





The global aerosol burdens normalized by the injection rate are shown in Fig. 3 as a function of injection rate. The normalized burden has units of time and can be considered the residence time of injected sulfur, which varies from 0.5 to 1.3 years for $SO_2$ injections and from 0.8 to 1.7 years for $AM-H_2SO_4$ injections. The CESM2 model shows longer residence times than the other models, consistent with its greater burdens. The $SO_2$ injections (Fig. 3, left panel) all show decreasing

residence time with increasing injection rate. This is consistent with other studies (Niemeier and Timmreck, 2015; Heckendorn et al., 2009) showing decreasing injection efficiency with increasing injection amount, which has been found to result from substantial increases in mean particle size and thus sedimentation rates. However, injections of $AM-H_2SO_4$ (Fig. 3, right panel) show increasing residence time with increasing injection for the CESM2 and MAECHAM5-HAM models. The $AM-H_2SO_4$ scenarios were designed to minimize the growth in average particle size as a function of injection rate, since

sulfur is added to each grid box as particles of approximately 0.1 μm radius which grow mainly by coagulation. In addition, aerosol heating in the tropical lower stratosphere increases the strength of the Brewer-Dobson circulation, resulting in greater lofting in the tropical stratosphere which, for accumulation mode and smaller particles, (Niemeier et al., 2020), can prolong the residence time for 25 Tg(S) yr$^{-1}$ injections relative to 5Tg(S) yr$^{-1}$ injections though the details of this process are model dependent and include changes in the QBO for the CESM2 and MAECHAM5-HAM models (Franke et al., 2021).

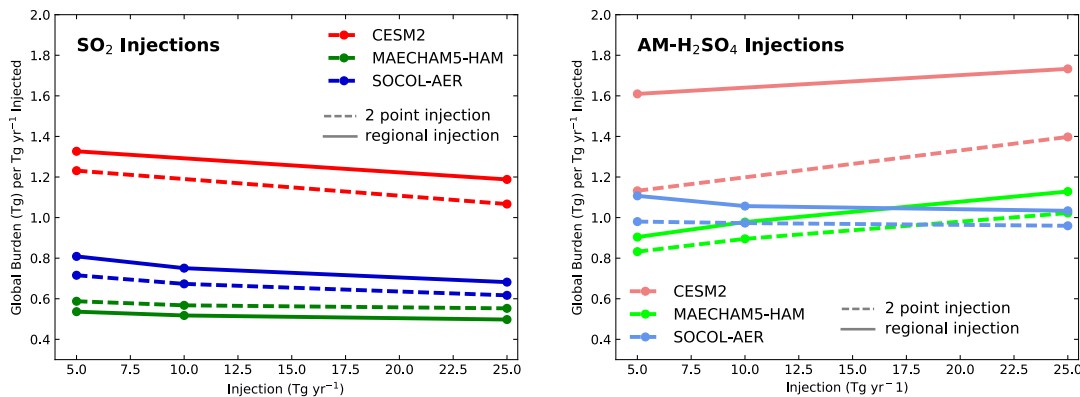

**Figure 3:** **Increase in global sulfate aerosol burden normalized by geoengineering injection rate in units of Tg(S) per (Tg(S) yr$^{-1}$) as a function of injection rate. The y-axis also represents the residence time in years of injected sulfur as aerosol. $SO_2$ injection (left panel) show decreasing residence time with increasing injection rate whereas $AM-H_2SO_4$ injections (right panel) show increasing residence time with increasing injection rate for the CESM2 and MAECHAM5-HAM models. Regional injections (solid lines)**
**show longer residence times than 2point injections (dashed lines) in most cases.**



Globally averaged effective wet radius ($R_{eff}$) at 60 hPa near the injection region and where these values maximize is shown in Fig. 4 (left panel), for background conditions and for injections of 5 and 25 Tg(S) yr$^{-1}$ of $SO_2$ or AM-$H_2SO_4$. The effective radius at 60 hPa after continuous injection of AM-$H_2SO_4$ results in $R_{eff}$ from 0.27 μm to 0.39 μm with 5 Tg(S) yr$^{-1}$ injections, whereas $SO_2$ injections yield $R_{eff}$ of 0.40 μm to 0.52 μm for the same annual sulfur injection. The $SO_2$ injections

consistently yield larger average particles than the AM-$H_2SO_4$ injections. As has been seen in other studies (English et al., 2012; Vattioni et al., 2019), injection of $SO_2$ results in substantial particle size growth since most of the injected sulfur condenses onto the larger existing particles or nucleates and preferentially coagulates onto the larger background particles. The assumed lognormal size of the input AM-$H_2SO_4$ particles in our scenarios is equivalent to a wet $R_{eff}$ of 0.18 μm (0.16 for MAECHAM5-HAM) and the additional particle growth is due to coagulation with both background and other injected

particles. For 5 Tg(S) yr$^{-1}$ injections of AM-$H_2SO_4$ particles, the resulting global averaged $R_{eff}$ at 60 hPa is within or smaller than the optimal $R_{eff}$ for scattering (blue band in the figure) whereas the $R_{eff}$ resulting from $SO_2$ injections is larger than optimal for scattering, particularly for regional injections. Increasing the injection rate from 5 Tg(S) yr$^{-1}$ to 25 Tg(S) yr$^{-1}$ results in larger mean particles in all cases, with the $SO_2$ injection scenarios responding more strongly than the AM-$H_2SO_4$ scenarios. The MAECHAM5-HAM model shows only small increases in $R_{eff}$ for the AM-$H_2SO_4$ scenarios as a function of

injection rate. The $SO_2$ injection scenarios all produce larger $R_{eff}$ with regional injections, while the AM-$H_2SO_4$ injection scenarios produce larger $R_{eff}$ with 2point injections.

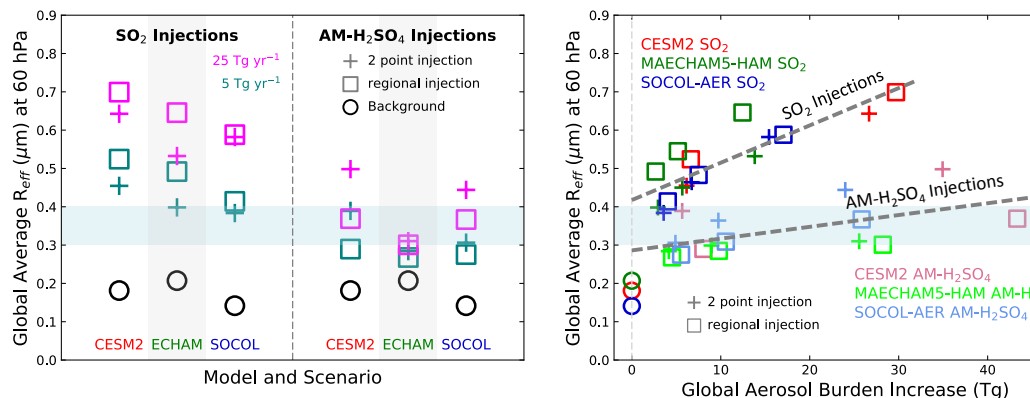

**Figure 4: Global average effective radius (mm) at 60 hPa (left panel) for the 3 models with $SO_2$ injections of 5 and 25 Tg(S)/yr (left**

**side) and with AM-$H_2SO_4$ injections of 5 and 25 Tg(S)/yr (right side). Scatter plot of global average effective radius (mm) at 60 hPa plotted against the global increase in aerosol burden in Tg(S) (right panel), with AM-$H_2SO_4$ injections shown with lighter symbols Regression lines are shown for both $SO_2$ ($R^2$=0.74) and AM-$H_2SO_4$ ($R^2$=0.34) injections. Injections into regions are shown by open squares, while injections at two points are shown with plus symbols, and for background conditions with open circles. The light**





**blue shaded region represents the optimal effective radius for scattering of solar radiation assuming a lognormal distribution and s between 1.1 and 1.8 (Dykema et al., 2016; Dykema private communication).**

Figure 4 (right panel) shows the same $R_{eff}$ parameter plotted against the increase in global aerosol burden for each case, including the 5, 10, and 25 Tg(S) yr$^{-1}$ scenarios. Linear regression lines are plotted for $SO_2$ injections and AM-$H_2SO_4$

injections (equal weighting of plotted points for all models), showing that the relationship between burden and $R_{eff}$ is close to linear though more steeply sloped for $SO_2$ injections. $R^2$ values are 0.74 for $SO_2$ injection and 0.34 for the AM-$H_2SO_4$ injections. $SO_2$ injection cases all lie above the AM-$H_2SO_4$ injection cases, with the latter yielding smaller $R_{eff}$ and larger burden for the same annual injection amount and even greater burdens with similar $R_{eff}$ values. The MAECHAM5-HAM model exhibits a somewhat flatter regression slope than the other models with AM-$H_2SO_4$ injections (smaller sensitivity of

$R_{eff}$ to burden) and an upward offset on the regression line with $SO_2$ for regional injections.

Particle size distributions for the 30° S-30° N region at 60 hPa are shown in Fig. 5 for $SO_2$ injections (left panel) and AM-$H_2SO_4$ injections (right panel) of 5 Tg(S) yr$^{-1}$. Note that the CESM2 model includes only 3 sulfate aerosol modes, omitting the nucleation mode seen in the MAECHAM5-HAM model. CESM2 generates new particles according to nucleation rates from Sitho et al. (2006), which are adjusted according to a parameterization from Kerminen and Kumala, (2002) and added

to the Aitken mode. The $SO_2$ injection scenarios result in an increase in nucleation mode particles relative to background levels (dotted lines in Fig. 5) in the MAECHAM5-HAM model and a decrease in the nucleation mode in the SOCOL-AER model. In these scenarios, the SOCOL-AER model results reflect a predominance of condensation, while the MAECHAM5-HAM results reflect a larger role for nucleation. An analysis with the SOCOL-AER model revealed that concentrations of nucleation mode particles are sensitive to the order of calculation of nucleation and condensation, the time splitting scheme,

and the time step, though this does not affect conclusions concerning the overall differences between $SO_2$ and AM-$H_2SO_4$ injections.

Large increases in the accumulation mode and coarse modes are seen for all models with $SO_2$ injection. The coarse modes have mode radii values ($R_g$) of 0.4 to 0.6 μm with MAECHAM5-HAM having the smallest coarse mode $R_g$ but also the smaller background distribution in the coarse mode. AM-$H_2SO_4$ injections decrease the nucleation mode and Aitken

mode particles as many of these particles are scavenged by the injected accumulation mode particles. Coagulation with background particles and with other injected particles moves the main particle size distribution from an $R_g$ of 0.1-0.12 μm upon injection to 0.2-0.3 μm for the CESM2 and SOCOL-AER models, while the MAECHAM5-HAM model retains a mode at 0.1 μm and also grows the coarse mode at 0.3 μm.



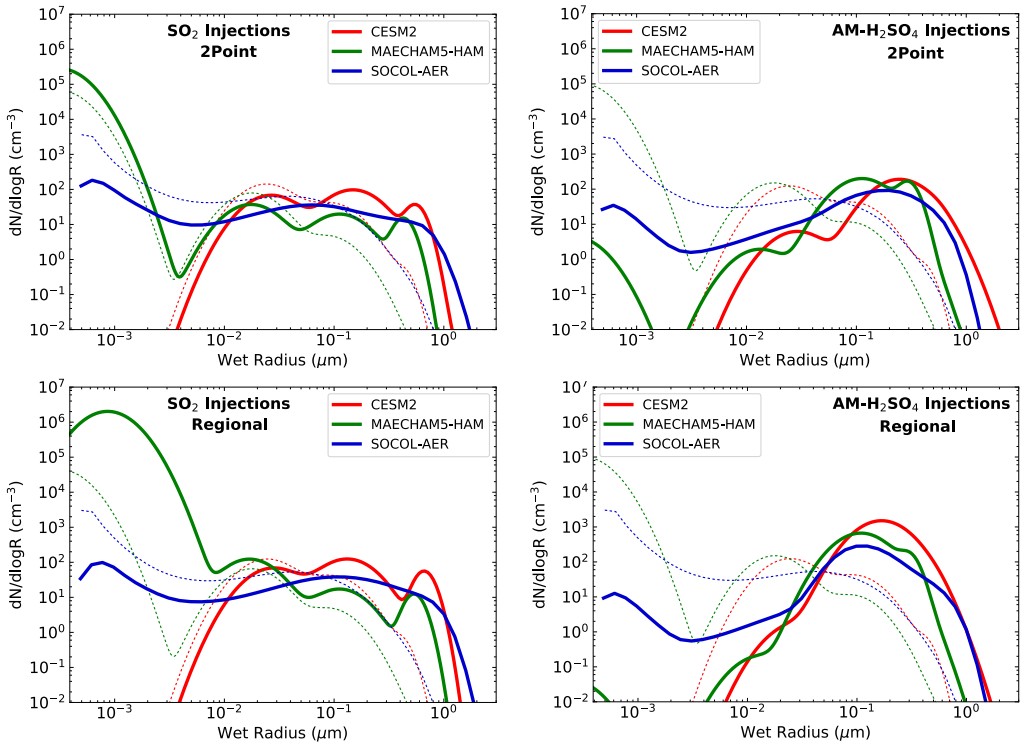

**Figure 5: Size distributions (dN/d$_{10}$logR, particles cm$^{-3}$ mm$^{-1}$) averaged between 30° S and 30° N at 60 hPa for the 3 models with**
5 **SO$_2$ (left panels) and AM-H$_2$SO$_4$ (right panels) injections of 5 Tg(S) yr$^{-1}$ and with point injections (top panels) and regional injections (bottom panels). Background size distributions are shown as dotted lines.**

The comparison between 2point and regional injections suggest the way aerosol microphysics drives differences between AM-H$_2$SO$_4$ and SO$_2$ injection scenarios (see Table 3). For AM-H$_2$SO$_4$, 2point injections produce larger R$_{eff}$ (Fig. 4) and smaller global burdens (Fig. 1) than regional injections. The regional AM-H$_2$SO$_4$ injection cases have size distributions
10 (see Fig. 4) which remain closer to their injected size distributions. We expect that injection of AM-H$_2$SO$_4$ into points increases the coalescence rate, driving the radius up and the lifetime down due to sedimentation of large particles. In contrast, SO$_2$ regional injections yield larger coarse mode particle sizes than the 2point injections, resulting in a larger R$_{eff}$. The 30-day conversion time from SO$_2$ to H$_2$SO$_4$ leads to H$_2$SO$_4$ condensation onto existing background particles that favors coarse mode growth. Small freshly nucleated particles in this scenario preferentially coagulate with the larger background





particles, also favouring coarse mode growth. Point injections of $SO_2$ are more likely to create locally high densities of nucleation mode particles that would coagulate among themselves, thus lowering the average size relative to regional injections. This effect is somewhat akin to injections into an aircraft plume, but on a very different scale. We expected this effect will be strongly dependent on model resolution which may partly explain the model discrepancies. Given our chosen

scenarios, some of the differences between regional and 2point injections are likely due to the interaction of dynamics with injection location – injections outside the tropics will less efficiently be transported in the upward branch of the Brewer-Dobson circulation, which could lead to faster stratospheric removal and lower global burdens for 2point injections.

**Table 3: Matrix Explaining 2point vs regional injection effects**

|  | **2Point Injection** | **Regional Injection** |
|---|---|---|
| **$SO_2$ Injection** | ➤ Similar to plume processing <br> ➤ More nucleation ⟹ more accumulation mode particles <br> ➤ Resolution-dependent impact expected | ➤ More coagulation with background or condensation onto background ⟹ more coarse mode particles |
| **AM-$H_2SO_4$ Injection** | ➤ More coagulation ⟹ more coarse mode particles <br> ➤ Resolution-dependent impact expected | ➤ Sizes remain closer to injected size, i.e. more accumulation mode |

### 3.2 Changes in radiative forcing and stratospheric temperature

Changes in the top of atmosphere (TOA) radiative forcing (RF) of shortwave (SW, solar) and longwave (LW, thermal) bands

combined are shown in Fig. 6 for our simulations with 5 and 25 Tg(S) yr$^{-1}$ injections under all sky conditions. RF changes range from –0.9 to –2.5 W m$^{-2}$ with $SO_2$ injections of 5 Tg(S) yr$^{-1}$ and from –1.6 to -3.8 W m$^{-2}$ with AM-$H_2SO_4$ injections of 5 Tg(S) yr$^{-1}$. For comparison, approximately –4 W m$^{-2}$ forcing would be needed to offset a doubling of $CO_2$ (Etminan et al., 2016). Intermodel differences encompass a factor of 3, and are larger than differences due to injection form (AM-$H_2SO_4$ vs $SO_2$) and pattern (2point vs regional) in the 5 Tg(S) yr$^{-1}$ case (left panel), but differences due to injection form are of a

similar magnitude as intermodal differences with 25 Tg(S) yr$^{-1}$ (right panel). The efficacy (RF reduction per Tg of sulfur injected annually) is shown in Fig. 7 as a function of injection rate. The efficacy is reduced with increasing injection rate for $SO_2$ injection scenarios, a consequence of the larger particles generated at high injection rates that both increase





sedimentation and decreases the shortwave scattering efficiency. Efficacy is also reduced with increasing injection rate for AM-$H_2SO_4$ injections with the SOCOL-AER model but stays roughly constant with injection rate in the CESM2 (2point injection only) and MAECHAM5-HAM models. Even though the normalized aerosol burden increases with increasing AM-$H_2SO_4$ injection rate for these models (Fig. 2), the RF efficacy is insensitive to injection rate, possibly due to the offsetting

5    effects of aerosol heating on circulation and more modest changes in particle size.

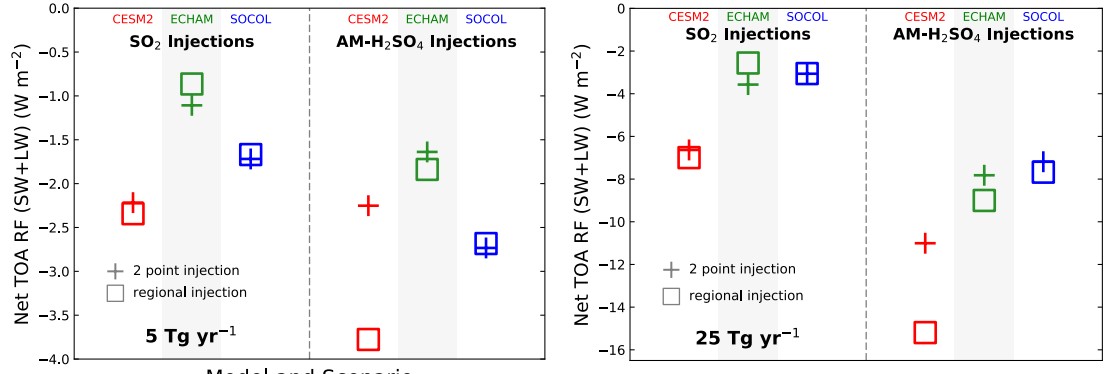

**Figure 6:** Globally averaged change in net top-of-atmosphere shortwave+longwave radiative forcing (W m$^{-2}$) due to geoengineering injection of (left panel) 5 Tg(S) yr$^{-1}$ and (right panel) 25 Tg(S) yr$^{-1}$. Colors and symbols as in Figure 1.





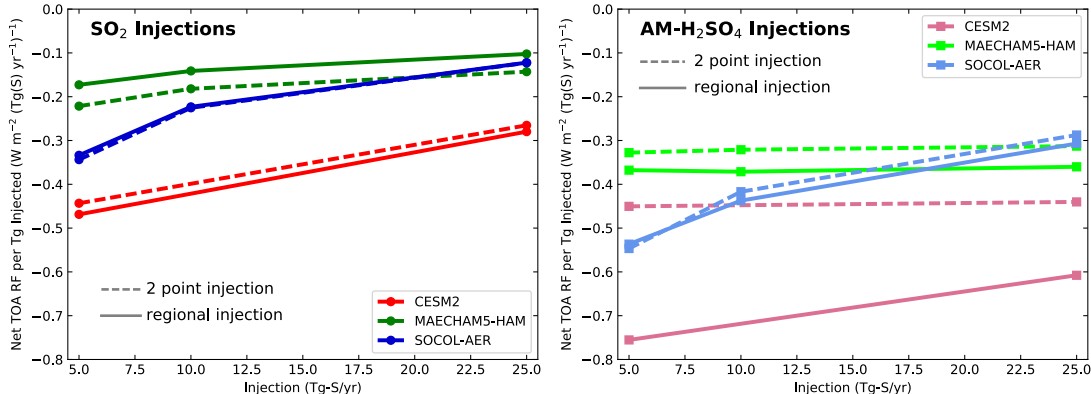

**Figure 7** Globally averaged change in net top-of-atmosphere shortwave+longwave radiative forcing (W m$^{-2}$) per unit annual injection (Tg(S) yr$^{-1}$) due to geoengineering injection as a function of injection rate for (left panel) SO$_2$ injections and (right panel) AM-H$_2$SO$_4$ injections.





Figure 8 shows a scatter plot of the change in SW+LW TOA RF plotted against the increase in global aerosol burden. The upper left insert of Fig. 8 expands the region from 0 to 18 Tg(S) burden increase. Linear regression lines for the $SO_2$ injections and $H_2SO_4$ injections are shown, with $R^2$ values greater than 0.9 in both cases, indicating that global burden is a good predictor of change in TOA RF. For the same burden increase, the AM-$H_2SO_4$ injection scenarios show somewhat

greater RF changes than the $SO_2$ injection scenarios, which we can attribute to a more optimal size distribution after AM-$H_2SO_4$ injections. Model differences in RF are, for the most part, contained in the differences in calculated burdens, with differences in size distributions accounting for additional variation.

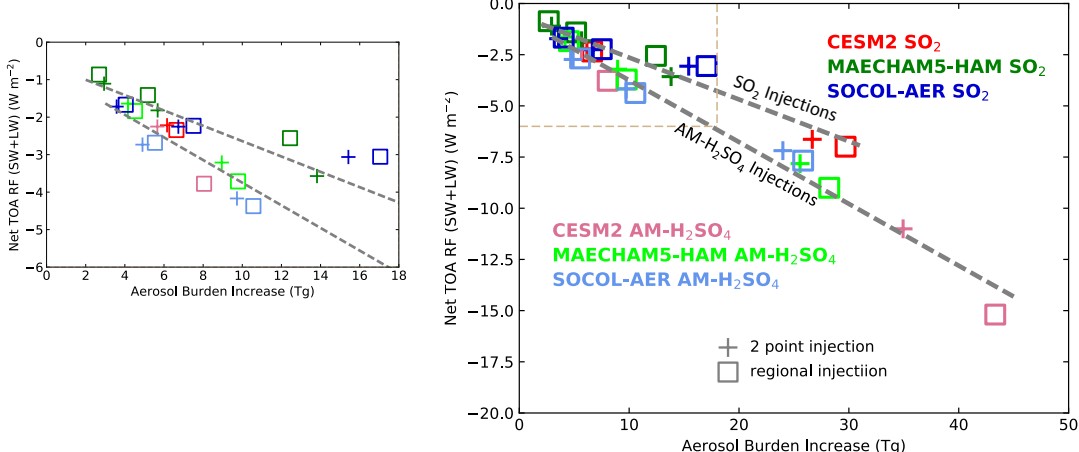

**Figure 8: Scatter plot of globally averaged net TOA shortwave+longwave radiative forcing (W m$^{-2}$) due to geoengineering injection relative to the corresponding increase in global aerosol burden (Tg(S)). The smaller figure is finer scale for the upper left corner of the main plot. Regression lines are shown for $SO_2$ injections ($R^2$=0.93) and for AM-$H_2SO_4$ injections ($R^2$=0.98).**

Figure 9 shows the latitudinal variation in net TOA SW+LW radiative forcing for both 5 and 25 Tg(S) yr$^{-1}$ injections.

The MAECHAM5-HAM model shows less variability of RF with latitude than the other models, while the CESM2 model shows much more variability. The SOCOL-AER model shows near-zero RF change, and sometimes positive RF change, in the high latitudes. The 2point injections in most cases produce greater RF change at mid and high latitudes as expected, while the regional injections produce a much greater impact on RF in the tropics. Compared to latitudinal changes in aerosol



burden (Fig. 3), changes in RF exhibit more small scale variability, a result of the high variability in tropospheric cloudiness.

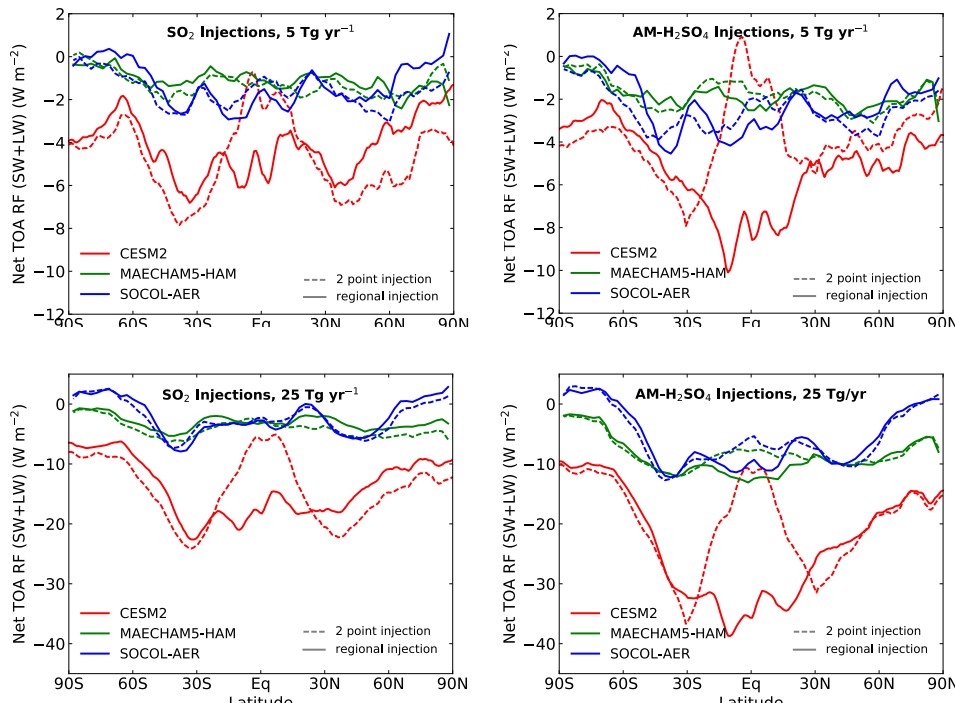

**Figure 9:** **Zonal mean net top-of-atmosphere shortwave+longwave radiative forcing with 5 Tg(S)/yr injections (top panels) and 25 Tg(S)/yr injections (bottom panels) as a function of latitude for SO$_2$ injections (left panels) and AM-H$_2$SO$_4$ injections (right panels). Regional injections are shown with solid lines and 2point injections with dashed lines.**

Next, we look at the vertical profiles of changes in tropical temperature (30° S-30° N) in Fig. 10. Sulfate aerosols absorb in the longwave and lead to atmospheric heating, which would lead to an enhancement in the Brewer-Dobson circulation and to increased transport of H$_2$O into the stratosphere. Increased stratospheric water vapor could lead to ozone losses via an enhanced HO$_x$ cycle in the upper stratosphere (Tilmes et al., 2018). Thus, aerosol heating in the tropical lower stratosphere is considered a serious risk of geoengineering. Previous studies indicate that such stratospheric changes could impact tropospheric climate (Simpson et al., 2019; Jiang et al., 2019), though we don't explore that here. Maximum model-calculated temperature changes in this region range from 1.7 K for the MAECHAM5-HAM model to 5.3 K for the CESM2 model with SO$_2$ injections of 5 Tg(S) yr$^{-1}$. With AM-H$_2$SO$_4$ injections of the same magnitude, model-calculated tropical temperature changes range from 2.1 to 6.4 K for the same two models. The SOCOL-AER model results are similar to the CESM2 model results with SO$_2$ injections of the same magnitude and injection pattern (2point vs region), though this


similarity does not extend to AM-H$_2$SO$_4$ injections. The larger temperature changes with AM-H$_2$SO$_4$ likely reflect the greater stratospheric burden for the same sulfur injection amount, a result seen in all three models (Fig.1).

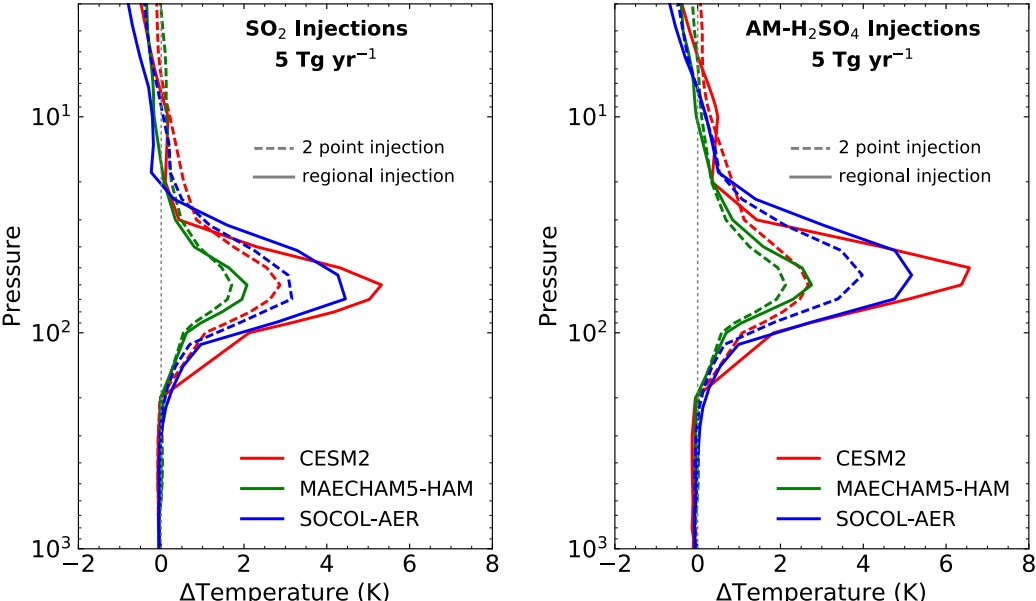

**Figure 10:** **Change in atmospheric temperature (K) averaged between 30°S and 30°N due to 5 Tg(S) yr$^{-1}$ of geoengineering injection as a function on height for (left panel) injections of SO$_2$, and (right panel) injections of AM-H$_2$SO$_4$.**

In Fig. 11 we look at the changes in H$_2$O and changes in temperature at 90 hPa in the tropics, shown as a scatter plot including injections of 5, 10, and 25 Tg(S) yr$^{-1}$. 90 hPa is close to the cold-point temperature which determines H$_2$O concentration entering the stratosphere, though the actual cold point could vary from model to model and from low to high injection rates. MAECHAM5-HAM results are not shown as H$_2$O was not calculated diagnostically in this model. Plotting H$_2$O against temperature shows that 90 hPa tropical water vapor mixing ratio is determined by 90 hPa tropical temperature, though with different relationships for the SOCOL-AER and CESM2 models. Since we plot these quantities averaged over time and spatial volumes, they do not follow the Clausius-Clapeyron equation but fall below it. Compared to control runs with H$_2$O values of about 4 ppmv in the tropics at this altitude, injections of 25 Tg(S) yr$^{-1}$ of SO$_2$ or AM-H$_2$SO$_4$ yield H$_2$O increases of 3.7 to 12 ppmv, which represents increases of factors of two to four. The SOCOL-AER model has larger increases in 90 hPa water vapor per degree of heating than the CESM2 model. As the H$_2$O vapor crossing the 90 hPa surface





in the tropics largely determines $H_2O$ concentrations throughout the stratosphere, this would be expected to significantly perturb stratospheric chemistry and ozone.

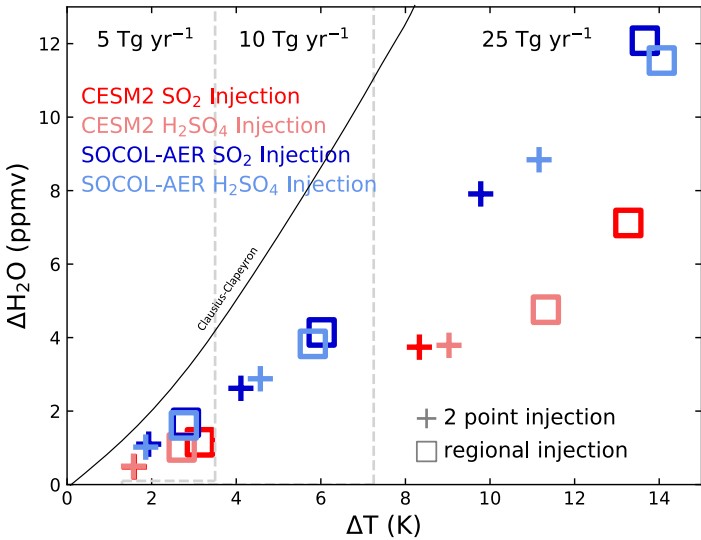

**Figure 11** Scatter plot of water vapor change (ppmv) relative to change in temperature (K) at 90 hPa and averaged between 30° S and 30° N for injections of 5, 10, and 25 Tg(S) $yr^{-1}$ of $SO_2$ or AM-$H_2SO_4$. Results from CESM2 and SOCOL-AER only are shown, as the MAECHAM5-HAM model used fixed $H_2O$.

### 3.3 Chemical changes

Increases in stratospheric water vapour concentration are expected to modify OH concentration in the stratosphere as well. However, OH chemistry is complex and $HO_x$ cycles interconnect with those of $NO_x$, $ClO_x$, and $BrO_x$. The CESM2 and SOCOL-AER models show significant increases in tropical OH concentration above 50 hPa, up to a 15% increase for SOCOL-AER and a 10% increase for CESM2 with 5 Tg(S) $yr^{-1}$ injection. These relative increases in OH are consistent with relative increases in $H_2O$ at 90 hPa. Analysis of our $SO_2$ injection scenarios shows that 7-10% of the additional global sulfur ($SO_2$+$SO_4$) burden remains as $SO_2$ in the SOCOL-AER model, 5-8% in the CESM2 model, and 20-22% for MAECHAM5-HAM. Derived residence times of the excess $SO_2$ burden (burden/$SO_2$ injection rate) are 18-28 days for SOCOL-AER, 25-37 days for CESM2, and 50-60 days for MAECHAM5-HAM. The long residence time for the MAECHAM5-HAM model



results from the prescribed OH field employed in these calculations. Residence times in all models change very little with injection rate from 5 to 25 Tg(S) yr$^{-1}$, indicating that OH is not depleted by these large continuous injections of $SO_2$, as this effect is counterbalanced by increases in stratospheric $H_2O$ due to heating of the tropical tropopause region.

Next, we evaluate the impacts on ozone, which are shown as zonal mean total ozone column (TOC) changes as a function of latitude in Fig.12 for the 5 Tg(S) yr$^{-1}$ case for the two models which include $HO_x$ chemistry (SOCOL-AER and CESM2). In agreement with previous studies (Pitari et al., 2014; see their Fig.14c), we find that $SO_2$ injections lead to a TOC decrease, which maximizes in mid and high latitudes. However we find larger polar losses in the southern hemisphere (20-30 DU) because of the larger sulfur injections in our study (5 Tg(S) yr$^{-1}$ vs 2.5 Tg(S yr$^{-1}$) in their study). Most of the column depletion occurs because of changes in the lower stratosphere, where the primary mechanism is $N_2O_5$ hydrolysis and

consequent formation of nitric acid ($HNO_3$), thus decreasing ozone loss due to $NO_x$ cycles and increasing it due to $ClO_x$ and $HO_x$ cycles. In addition, chlorine is activated via heterogeneous reaction of $ClONO_2$ and $H_2O$ on stratospheric aerosols, contributing to most of the ozone depletion in polar latitudes. Both chemical pathways are enhanced via enhanced aerosol burden and consequent larger surface area density (SAD) values. The impact of chlorine on ozone is a function of the simulation year (2040) and future projection chosen, with our 2040 simulations containing 2.4 ppbv of total chlorine.

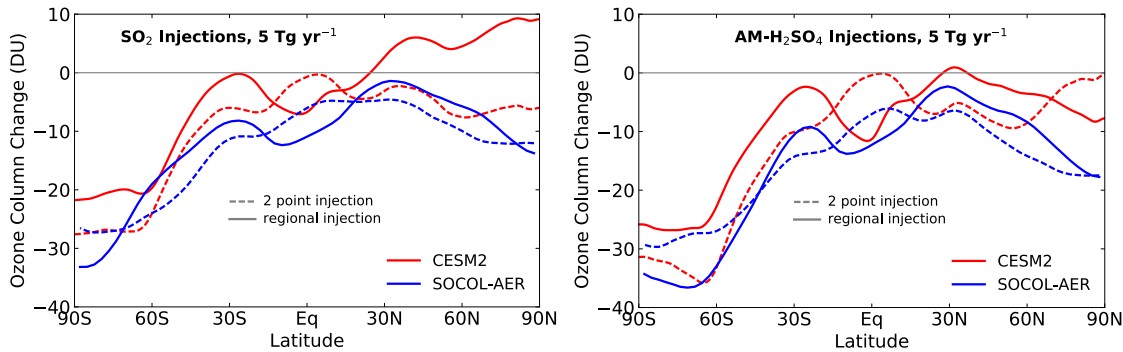

**Figure 12:** Average change in column ozone (Dobson units) due to geoengineering injection as a function of latitude for (left panel) 5 Tg(S) yr$^{-1}$ of $SO_2$ injection and (right panel) 5 Tg(S) yr$^{-1}$ of AM-$H_2SO_4$ injection.

Ozone decreases in the lower stratosphere are partly offset by ozone increases in the middle stratosphere (10-50 hPa), due to weakened $NO_x$ depletion cycles, in agreement with other studies (Heckendorn et al., 2009). The CESM2 model, in

fact, shows increases in total ozone poleward of 30°N with regional injections of $SO_2$. AM-$H_2SO_4$ injections lead to a very similar TOC pattern as $SO_2$ injections in both models, although depletions are slightly larger (by 10-20%) with AM-$H_2SO_4$ due to larger sulfate aerosol burdens (Fig.1) with smaller mean particle size and consequently larger SAD throughout the stratosphere. Hence, more surface area is available for heterogeneous reactions, leading to larger ozone depletion in the case of AM-$H_2SO_4$ rather than $SO_2$ injections, consistent with previous findings (Vattoni et al., 2019). Note that while the



CESM2 model has larger increases in aerosol burden than the SOCOL-AER model, it nevertheless shows smaller changes in total ozone.

Stratospheric sulfate geoengineering can have a strong impact on Arctic and Antarctic ozone depletion (Tilmes et al., 2009). However, this effect is generally less severe in the Arctic, and is strongly modulated by inter-annual variations in the

polar vortex strength. Both models produce Antarctic ozone depletion of 20-35 DU, while the Arctic shows smaller depletions (up to 18 DU) for SOCOL-AER. The CESM2 model shows minimal Arctic ozone depletion (up to 8 DU) for most cases but an increase in Arctic column ozone with $SO_2$ regional injections where positive changes in the middle stratosphere dominate negative changes in the lower stratosphere. Longer simulations than those considered here (8 years) would be needed to robustly detect dynamical and chemical effects on the Arctic stratosphere. Lastly, little sensitivity of the

ozone response to the injection strategies (regional vs 2point injections) is seen, possibly due to opposing effects of injection pattern on aerosol burden in high and low latitudes, as seen in Fig. 2 (i.e., 2points injections lead to larger polar burdens, whereas regional injections lead to larger tropical burdens).

Figure 13 shows the correlation between global ozone change and global aerosol burden (left panel) or net TOA RF (right panel). Changes in total ozone column are a mix of both positive and negative local changes, and the global average includes large depletions in the Antarctic and small to moderate depletions elsewhere. The correlations with burden and net

includes large depletions in the Antarctic and small to moderate depletions elsewhere. The correlations with burden and net TOA RF are fairly compact for the SOCOL-AER model ($R^2$=0.95 for burden, 0.90 for RF) but less so for the CESM2 model ($R^2$=0.58 for burden, 0.72 for RF) which is much more sensitive to injection form and location. Regression lines for the SOCOL-AER model are much steeper than for the CESM2 model, indicating different ozone sensitivities in the two models. A -4 W m$^{-2}$ change in net TOA RF corresponds to global ozone changes of from -1.5% to -5% among our 2 models and four

injection scenarios, indicating large model uncertainties in ozone response in SSG.

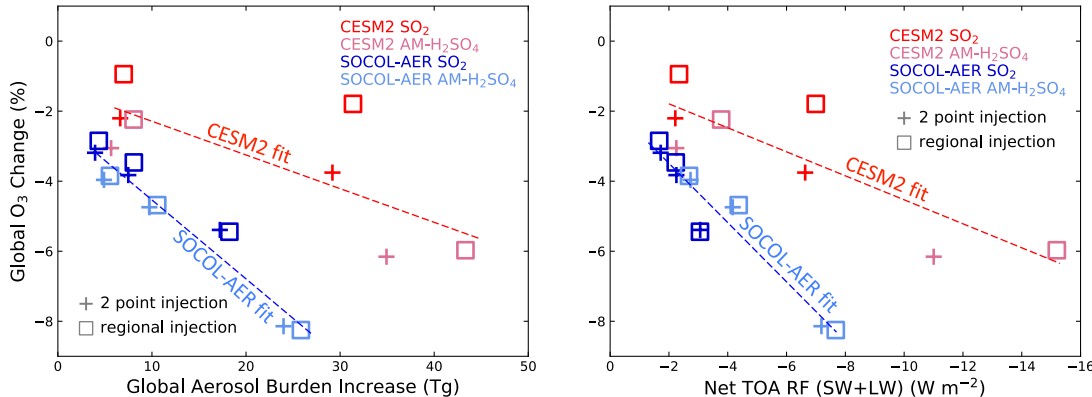

**Figure 13:** Global average change in column ozone (%) due to geoengineering injection plotted against (left panel) the global average aerosol burden increase (Tg(S)), and (right panel) the average reduction in SW+LW radiative forcing (W m$^{-2}$). Regression lines are shown for each model with R$^2$ values of 0.95 and 0.90 for SOCOL-AER (burden and RF, respectively) and of 0.58 and 0.72 for CESM2.

### 4 Summary and conclusion

Most obviously, the fact that all three models show that use of AM-H$_2$SO$_4$ particles can aid in controlling large-scale particle size distribution strengthen the case this method might be useful for sulfate aerosol SSG. Improved control of particle size can, in turn, allow use of less sulfur to achieve the same radiative forcing, or could allow higher levels of radiative forcing with less nonlinear saturation. Our three-model intercomparison increases the confidence in this general result while simultaneously demonstrating the significant uncertainty that arises from differences in model dynamics and model treatment of aerosol microphysics and chemistry. We note first there are large inter-model differences in both absolute quantities such as aerosol burden and radiative forcing and in derivative quantities such as aerosol lifetime and the change in radiative forcing with injection rate. Nevertheless, the intra-model differences in the impact of SO$_2$ vs AM-H$_2$SO$_4$ show systematic agreement among the models. The intermodel differences in radiative forcing and R$_{eff}$ are consistent with the intermodel differences in aerosol burden as diagnosed by the compact relationships among these quantities. And the significant difference in correlation slopes between SO$_2$ and AM-H$_2$SO$_4$ injection scenarios indicates that the two injection forms have different radiative efficacy related to their size distributions.





Perhaps the most interesting result is the systematic differences in radiative efficacy achieved over the 2 x 2 matrix of cases formed by the choice of $SO_2$ vs $AM-H_2SO_4$ and regional vs 2point injections, which is summarized in Table 3. These results hint at the limitations that come from the unresolved spatial scales between the injection plume and the model grid boxes, limitations that are common to these three models along with all other global models that have been used for studying

SSG.  For $AM-H_2SO_4$, 2point injections produce larger particles and lower radiative efficacies than regional injections. This is consistent with the expectation that point injections will drive up coagulation rates producing inefficiently large particles.

Results are less consistent for $SO_2$. All models find that 2point injections decrease particle size relative to regional injections, but the models do not agree on the sign of the difference in radiative efficacy between 2point and regional injections. The decrease in particle size may be due to point injections in the GCM providing the same physical mechanism

that is simulated in plume models in which new accumulation-mode particles are created by high densities of $SO_2$ and subsequently $H_2SO_4$ gas and locally high densities of nucleation mode particles. But while the physical mechanism is similar, the length and time scales differ by many orders of magnitude as the plume level simulations show that accumulation-mode particles would be formed from $H_2SO_4$ condensation in an aircraft plume on a timescale of minutes in a plume that has a horizontal length scale of 10's of meters. If $SO_2$ were actually injected from aircraft, it would form high

aspect ratio plumes that observations suggest remain coherent for timescales of at least a week, and these plumes might have chemical and microphysical dynamics that are quite different from those simulated on the scale of a typical model grid box.

The most surprising and puzzling result is the increase in aerosol burden per unit $AM-H_2SO_4$ injected with increasing injection rate for two of the models (CESM2 and MAECHAM5-HAM). This may reflect the balance between increased tropical upwelling due to aerosol heating and increased sedimentation as a function of particle size and may be influenced by

interactive changes in the QBO. This result, however, may depend on the initial size distribution of the $AM-H_2SO_4$ input to the GCMs as well as details of the model's resolution and transport processes and their interaction with aerosol microphysics. The SOCOL-AER model, which employs a sectional aerosol scheme, may remove the largest particles by sedimentation more efficiently than the modal schemes employed in the other models, thus leading to a decrease in aerosol burden per unit $AM-H_2SO_4$ injected with increasing injection rate in this model.

We have examined two side effects of geoengineering in this study:  changes in lower stratospheric tropical temperature and changes in ozone. The use of $AM-H_2SO_4$ injections rather than $SO_2$ injections does not ameliorate these side effects when comparing equal injection amounts by sulfur weight. However, Fig. 8 shows that similar net RF could be achieved with a ~35% smaller aerosol burden with $AM-H_2SO_4$ than with $SO_2$, which likely reduces these side effects per unit of RF.

This study is a step towards systematic study of the effectiveness and limits of using $AM-H_2SO_4$ to influence the particle

size distribution during a hypothetical deployment of SSG. Yet it is only one small step, and our results are subject to significant limitations including:

• The treatment of aerosol microphysics is inconsistent in that two of the models used a modal scheme (CESM2 and MAECHAM5-HAM) and one of them used a sectional scheme (SOCOL-AER). And the size boundary between





accumulation mode and coarse mode differs between the two modal models. We find these model differences to be especially problematic in the large size tail of the size distribution that most influences the overall sedimentation rate.

• Results undoubtedly depend on resolution, and resolution varied significantly across models used here (see Table 2). CESM2 has a much finer horizontal resolution than the other two models, and the SOCOL-AER model was noticeably

coarser in vertical resolution. The coarse vertical resolution of SOCOL-AER precluded interactive QBOs that are known to influence SSG simulation results (Niemeier and Schmidt 2017; Franke et al., 2021).

• Our results on AM-H$_2$SO$_4$ aerosols depend on the aerosol size distribution we provided as input to the models. This distribution is intended to represent the distribution that would arise following processing within an aircraft plume and dispersal of that plume into a well-mixed grid box. But that process is not resolved in these models and is deeply uncertain.

We do not know how that distribution would depend on the specifics of injection including location, local temperature and turbulence, injection rate and aircraft characteristics, and the aerosol size distribution in the background air. Finally, this distribution implicitly assumes that the AM- H$_2$SO$_4$ is introduced by an aircraft plume, but other deployment methods have been considered such as tethered balloons.

• Observations (Murphy et al., 1998) suggest that the actual composition of lower stratospheric aqueous aerosols is not

purely sulfuric acid but may contain a significant amount of secondary organics and minor amounts of meteoritic and other materials. The presence of secondary organic aerosols may be expected to change both chemical and optical properties of sulfate aerosols. These processes are not accounted for in any of our models and are likely to vary spatially and seasonally.

• All these models may suffer from limitations in stratospheric dynamics and mixing (Linz et al., 2017; Niemeier et al, 2020; Dietmueller et al., 2018). For example, we expect substantial differences between mixing dynamical processes in the

relatively low vertical resolution SOCOL-AER and the high-resolution CESM2.

Improved understanding of the effectiveness of stratospheric sulfate injection and the role of plume-scale formation of accumulation mode particles may require use of modelling methods such as plume-in-grid or adaptive mesh to better capture the multi-scale problem from injection plume to the global circulation. Nonlinear interactions between aerosols and chemical species need to be explored across spatial scales. Small-scale field studies of aerosol dispersion and growth in the

stratosphere under various conditions could reduce uncertainty. However, uncertainties will remain in predicting the performance and impact of any solar geoengineering technology.

Data available at: https://dataverse.harvard.edu/dataverse/AM-H2SO4_Intercompare_Data

**Author Contributions**

DK originally proposed the study. All authors discussed the idea of the study and mutually agreed to the model boundary conditions and sulfate injection parameters. DV performed the CESM2 simulations. HF and UN performed the





MAECHAM5-HAM simulations. SV and GC performed the SOCOL-AER simulations. Significant scientific guidance on the overall project was provided by TP. DW produced the plots and performed the model comparisons with input from other authors. DW drafted the majority of the manuscript with DK and GC drafting sections, and all authors contributing to the final manuscript.

## 5  Acknowledgments

The Harvard University Solar Geoengineering Research Program supported DW and DK. Support for DV was provided by the Atkinson Center for a Sustainable Future at Cornell University. Ulrike Niemeier obtained support from the German DFG-funded Research Unit VollImpact FOR2820 grant no. 398006378. MAECHAM5-HAM simulations used resources of the Deutsches Klimarechenzentrum (DKRZ) granted by its Scientific Steering Committee (WLA) under project ID bm0550. HF would like to thank UN and Stefan Bühler (University of Hamburg, Germany) for enabling him to work on this very exciting master's thesis project and for excellent supervision. DV would like to thank Simone Tilmes, Michael J. Mills and Jadwiga Richter for assistance with running CESM2(WACCM), and the high-performance computing support from Cheyenne (https://doi.org/10.5065/%20D6RX99HX) provided by NCAR's Computational and Information Systems Laboratory, sponsored by the National Science Foundation. SV and DW thank Eric J. Klobas of Harvard University for assistance compiling and debugging SOCOL-AER on the Harvard computer system. GC and SV would like to thank Andrea Stenke and Timofei Sukhodolov for technical support with SOCOL-AER and discussion of the results.

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
