# Peer review of "An Aerosol-Climate Model Intercomparison of Stratospheric Solar Geoengineering by Injection of SO2 or Accumulation-Mode Sulfuric Acid Aerosols"

_Atmospheric Chemistry and Physics, 2021_

## Referee Comment (RC2)

Review of manuscript "A Model Intercomparison of Stratospheric Solar Geoengineering by Accumulation-Mode Sulfate Aerosols" by Debra Weisenstein et al.

This manuscript presents results from a model intercomparison comparing interactive stratospheric aerosol simulations within co-ordinated multi-model experiments to explore the global dispersion and radiative forcing that would result from a continuous source of sulphur dioxide or accumulation mode sulphate aerosol particles with two different emissions scenarios: one emitting only at 30N and 30S, the other as a constant source between 30S and 30N.

The intercomparison compares results from 3 different interactive stratospheric aerosol models (WACCM-MAM3, MAECHAM5-HAM and SOCOL-AER), and represents a potentially very interesting contribution to understand the predictions from the models, each having differing sophistication in their aerosol modules, and in the vertical and horizontal resolution of the GCM's advection.

Whilst the results are interesting, and certainly will be publishable in a revised form, the aim and design of the model experiments are surprisingly poorly described, and the Introduction and interpretation need to include some discussion also of the tropical stratospheric reservoir, in relation to the differences between the two scenarios.

In several places the manuscript has unscientific language and vague statements that need to be changed to terms more appropriate to a journal article. For example "may produce overly large aerosols" (page 1, line 12) and "unfavorable aerosol size distributions" (page 2, line 20) and "Our aerosol size distribution" (page 3, line 14) are clearly subjective terms that need to be better phrased to communicate the issues involved.

There are also a few places where the wording is poor, for example "These limits might be addressed" (page 2, line 25), "some of these limits may be addressed by altering the size distribution of sulfate aerosol" (page 2, lines 26-27).  The authors are clearly aware that these issues are at the heart of the science to understand the efficacy and risk of a hypothesised large-scale injection of precursor gas of idealised particle for solar radiation management. Referring to "altering the size distribution of the sulfate aerosol" grossly simplifies the complex interplay of processes involved -- and the wording needs to communicate consistently with an awareness of these issues.

I also find it very surprising that, in this initial version of the manuscript, the authors have not adequately explained the rationale for the very interesting model experiments they are presenting results from.

The two "injection scenarios" presented in the paper: 1) emitting continuously at two sites at 30N and 30S, and 2) emitting continuously throughout all latitudes between 30N and 30S, not surprisingly cause very different enhancements to the stratospheric aerosol layer (as is clearly seen in Figure 2). The 30N and 30S two-site scenario causes the stratospheric aerosol layer enhancement to be almost exclusively in mid- and high-latitudes, with only a very minor elevation in stratospheric aerosol optical depth in the tropics.

The main reason for this is of course that the two-site injection scenario emits SO2/particles entirely outside the tropical stratospheric reservoir (between 20S and 20N). It is well established (e.g. Dyer, 1974) that the residence time for volcanic aerosol clouds formed in the tropical stratosphere are much longer (around 2 years) than for eruptions forming stratospheric aerosol clouds in the mid-latitudes.  The reason is the continuing tropical upwelling and the transport barrier at the edge of the tropical pipe, and analysis of satellite measurements in 1991-1992 show the effect for example on the dispersion of the Pinatubo aerosol cloud (see Grant et al., 1996

for example). There needs to at least be sentence briefly mentioning the tropical stratospheric reservoir in the Introduction, and some discussion of the seasonal cycle of the Brewer-Dobson circulation (e.g. as set out originally by Dyer et al., 1968 for the Agung aerosol cloud).

Just to be clear, my review is not saying these results are not interesting, the results are indeed very interesting --- and this is a laudible effort to have a set of experiments to better understand any differences in predictions with the models -- but there needs to be a much clearer explanation of the rationale for why these scenarios were chosen.

It's implicit in the text that the 30N and 30S case might represent a limited 2-site injection strategy, but it needs to be made clear that the two scenarios are not really comparable, and that our understanding of stratospheric circulation would clearly mean that the 2-site 30N and 30S injection scenario would give a mid-latitude focussed stratospheric aerosol forcing, whereas the 30S-to-30N area-source scenario is presumably designed to give a more evenly-spread stratospheric aerosol layer enhancement, with then substantial radiative forcing also in the tropics.

I'm also recommending the authors consider changing the title, because the term "Solar Geoengineering by Accumulation-Mode Sulfate Aerosols" is not consistent with what the authors state that particular model experiment is representing.

Firstly, the models ran separate simulations with continuous emission of SO2, in addition to the experiment with the particle source, so the experiments are to explore also the injection of SO2, in addition to direct injection of particles. So the title should either state that both are carried out, or else just give a more general summary-term there.

Secondly, the manuscript states (page 3, lines 14-17) that the particle-injection experiment is designed to represent particle sizes that would occur at the grid-scale of the large scale models following injection of SO3 or H2SO4 from high altitude aircraft, a localised plume subsequently generating a source of accumulation mode particles at the grid-scale of the GCMs.

The text states "Our aerosol size distribution is consistent with Pierce et al. (2010) and Benduhn et al. (2016) who modelled plume microphysics and found that injection rate could be adjusted to produce sulfate aerosol size distrihbutions in the 0.1-0.15 micron radius size range."

I'd say first that I'm not sure either of those two first authors would argue that one can simply adjust the injection rate to produce the desired size distribution. I would expect that both would explain that there would be a substantial variability in the size distribution generated as the plume subsequently entrains into, and becomes mixed with the surrouding ambient air.

So I'd argue that the text "could be adjusted to produce" is not really adequately representing the eventual variability in sizes that would result there.

That said, I accept that a large diversity range for particle size is given (0.1 to 0.15 microns). It's again a case of the wording not adequately communicating the issues involved.

In my specific comments below, I'm recommending the authors consider using the terminology "sub-grid-scale sulphate emission" rather than "accumulation-mode particle emission", or as an alterntive they could actually explain the rationale of the experiment is to represent a proxy for an idealised particle source, deliberately designed to produce particles at a particular desired size.

In light of the likely large variability in particle sizes that continued gaseous emission of

SO3 or H2SO4 would cause, to me it is actually this engineered particle-emission scenario that these controlled size-distribution experiments are representing.

The other similar terminology issue I identify in my specific comments, is that the authors seem to use both the acronym "SRM" for solar radiation management, and also use the acronym "SSG" for stratospheric solar geoengineering.

In my view, the paper needs to be consistent in either using SRM or SSG, but not both.

My recommendation would be to use the acronym SRM, since the acronym SSG is often used for "scientific steering group", and SRM is also (in my mind) the more established term.

I'm suggesting the title should also be clear these are interactive stratospheric aerosol simulations being intercompared, with my suggestion being to change the title from

"A model intercomparison of stratospheric solar geoengineering by accumulation-mode sulfate aerosols"

instead to something like:

"A co-ordinated intercomparison of interactive stratospheric aerosol model experiments for hypothesised scenarios of solar radiation management by sulfate aerosols"

I provide below a list of specific comments I am asking the authors to address, and with these comments requesting a change in the tone of the narrative of the manuscript, my review then finds major revisions are needed.

The authors may find it relatively easy however to make these changes -- with the Figures, and much of the results section is in good shape, requiring only minor revisions.

Specific comments:
* * *
1) Page 1 -- lines 1-2 -- Further to the comments above, I strongly recommend the authors consider using a different term than "solar geoengineering by accumulation-mode sulfate aerosols". The optical depth from the stratospheric aerosol layer mainly comes from sulfate aerosols in the accumulation mode size range, and the forcing from any geoengineered enhancement to the stratospheric aerosol layer would be caused by particles in the accumulation mode part of of the size spectrum.   So using the precursor term "accumulation-mode" prior to the "sulfate aerosols" is not really a useful descriptor of the effect.

I realise that one of the co-ordinated multi-model experiments involves each model adding a continued source of sulfate aerosol particles at a particular constant size distribution (in the accumulation mode size range) but that term is then referring to some specifics of the design of the model experiment.

Remember that the residence time of particles in the stratosphere is months to years and the resulting size distribution from a continued emissions is rather a response to that source of particles, and the microphysical and dynamical processes likely mean the resulting size distribution may differ substantially from that within a localised primary emission. That does not necessarily rule out devising an source of particles engineered to achieve a particular resulting size distribution.  But a terminology referring simply

to "solar geoengineering by accumulation-mode sulfate aerosols" could lead to some readers inferring too simplified a relationship between the size distribution at particle emission and the evolving size distribution of the resulting dispersed aerosol cloud.

The authors refer to Pierce et al. (2010) and whilst the 2D-AER interactive stratospheric aerosol simulations give a reasonable assessment for the progression of the geoengineered aerosol cloud, the dilution of the initial plume and its subsequent evolution of the size distribution of the dispersed aerosol within the stratospheric dynamics of a higher resolution 3D GCM may well have given differing result.

As I say, I am not at all underplaying the value of these model experiments, which could well help to shed light on some of these issues, but I strongly advise the authors use a different terminology for the mechanism the model experiments are investigating.

Within the article, the authors need to be clearer whether these experiments really are representing a scenario of injected H2SO4 vapour. With the resulting plume rapidly nucleating particles to form a source of new particles that progress to be large enough to scatter incoming solar radiation.

The current model experiments do not really represent that situation, because there would certainly be greater variability in the size distribution in that case of H2SO4 vapour emission.

Rather I would argue these experiments mimic a situation where particles are emitted with a controlled size distribution, the particles deliberately engineered to achieve a certain subsequent response within the stratospheric aerosol layer.

My recommendation in this first comment is to change to a more general title something like:

"A co-ordinated intercomparison of interactive stratospheric aerosol model experiments for hypothesised scenarios of solar radiation management by sulfate aerosols"

That's partly because the experiments are not restricted to only assess an emitted source of particles, they also assess the models' response to emitted SO2.  In light also of the potentially large variations in particle size distribution that would result, to simply tag the approach as "Accumulation mode particle geoengineering" is not appropriate (in my opinion).

As per the subsequent specific comments, within the article, I can understand there is a benefit to referring to the effect from the strategy (in that it is specifically introducing accumulation mode particles into the models), but still I'm recommending the authors use a different terminology than "AM-H2SO4 geoengineering".

Global aerosol microphysics modellers may tend to use the term "sub-grid scale particle formation" or "primary sulphate emission" for this approach, with the former being much preferred to the latter by experts. And in my comments then I advise to use the term "sub-grid scale particle formation model experiments" or similar as the alternative term.

2) Page 1 -- Abstract, line 11 -- Suggest to insert "tended to focus on" rather than simply "focussed on", and rather than the somewhat vague term "Analyses", be clear you're referring to interactive stratospheric aerosol model analyses". In fact probably better to use "studies" rather than "analyses".

3) Page 1 -- Abstract, lines 11-12 -- the 2nd half of this 1st sentence then refers to climate models (whereas I think the first half refers to interactive stratospheric aerosol models). I think I understand what the authors mean when they say the climate model experiments "have assumed injection of SO2", but that could confuse some readers, because the majority of climate models do not tend to use their interactive aerosol modules for stratospheric aerosol, and therefore do not tend to represent injection of SO2 at all.

I think what the authors mean is that the model experiments tend to be designed to represent a scenario of imposing radiative effects consistent with best estimates of what could be expected from continued injection of SO2.

I suggest to change "have assumed injection of SO2" to "are based on scenarios aimed to represent the effects from continued SO2 injection". Or similar.

4) Page 1 -- Abstract, line 12 -- As per my general comments above, "may produce overly large aerosols" is obviously unscientific language. Also, that particles grow larger with increased SO2 is a scientific fact, with then use of the word "Yet" not good grammar.

It's an important point the authors are making, but this should be stated in an objective way, whereas the precursor word "Yet" suggests the authors consider it somehow unfortunate or undesirable.

Suggest "It is well established (e.g. Pinto et al., 1989) that greater emission of SO2 leads to larger sulphate aerosol particles, with shorter residence time in the stratosphere."

5) Page 1 -- Abstract, line 13 -- I think changing "new" to "additional" changes to a more accurate representation, to ensure authors do not mistakenly infer that particles form immediately at accumulation mode sizes (but rather grow from an initially smaller Aitken mode sizes) in this scenario of aircraft injection of SO3 or H2SO4.

This is an example of where I think the simplified term "geoengineering by accumulation mode sulphate" might only increase the probability of an incorrect inference in that respect.

I therefore strongly suggest the authors delete "AM-H2SO4", as the acronym similarly will tend to embed an increased likelihood of that over-simplified perception of the progression of the microphysical and dynamical processes involved.

The term "nudged" is also not appropriate in this context, tending to over-simplify the response of the stratospheric aerosol layer.

I'd suggest to re-word to "Some studies have explored whether a stream of very small particles can be generated by injecting H2SO4 vapour rather than SO2, potentially then leading to longer-lived aerosol paricles for a given sulphur injection rate."

Introducing a specific delivery mechanism seems un-necessary, and my suggested re-wording then also keeps the point more general than that specific situation of aircraft injection.

6) Page 1 -- Abstract, line 15 -- For the reasons given earlier, please change the terminology "AM-H2SO4 injection" to refer to the specifics of the model experiments rather than an apparently more general "type of geoengineering".

As explained in my comments above, I'm suggesting to use the term "sub-grid scale source of particles" as the descriptor, referring then to the specifics of the model experimetns, with also an acronym then not required in this case.

I suggest then to change this sentence to instead say "Whereas GeoMIP has included experiments to intercompare SO2 injection scenarios, the results here are the first multi-model intercomparison of the effects from a sub-grid scale source of sulphate aerosol. Or something like this.

With the subsequent sentence referring to GeoMIP, suggest to reserve the statement of "first" for after the sentence referring to GeoMIP. The scope of that sentence can be made more general by changing "We compare three models" to "A co-ordinated multi-model experiment designed to represent this SO3- or H2SO4-driven geoengineering scenario was carried out with 3 interactive stratospheric aerosol models:". The word "coordinated" can then be deleted later in the sentence.

7) Page 1 -- Abstract, line 24 -- Further to my general comments above, the term "sensitivity to injection pattern" is not an adequate description of the two experiments, the two-site experiment resulting in a midlatitude-focused forcing with little enhancement to the tropical stratospheric reservoir.  The word "sensitivity" suggests a slight change whereas these two alternative representations of the geoengineering enhancement are much more substantially different.  Better to actually crystallise in the reader's mind what the two alternative scenarios represent -- a mid-latitude-focussed forcing (presumably designed to avoid perturbing climate-sensitive regions in the tropics?) and an evenly distributed injection rate across the tropics and mid-latitudes.

Suggest then to change the sentence beginning "We explore the sensitivity to injection pattern" to "Simulations with two scenarios were designed to compare a two-site injection focussed to force only the mid-latitudes, with a more evenly distributed geoengineering forcing, each run with both SO2 and sub-grid particle experiments." or something like this.

The "and find opposite impacts" is explaining the results, and should be explained in a separate sentence, changing "and find opposite" to "We find opposite" or similar.

8) Page 2 -- Introduction, line 2 -- insert "long-wave" before "radiative forcing" and change "from the rise in CO2" to "from increased CO2 concentrations".  The reason here is to keep in the reader's mind that emissions are not necessarily the same as concentrations.

9) Page 2 -- Introduction, line 3 -- "Despite the complexity" -- it's not corect to say "Despite" here and I'd argue it's more "because of the complexity" that these models are needing to be used to try to predict how the overall system responds given the complex interactions and feedbacks.

Suggest to change "Despite the" to "In light of the".

10) Page 2 -- Introduction, line 4 -- "solar radiation management (SRM) is being studied". It's not really the solar radiation management itself that is being studied -- it's the effects from hypothesized solar radiation management (whether that be the responses of the stratospheric aerosol and ozone layers or the surface response to climate and the hydrological cycle).

Suggest to insert "the effects from hypothesized" after "carried risks".

11) Page 2 -- line 5 -- since the word "climate" is used later in the sentence (and with the change above also earlier in the sentence) change "climate models" to "earth system models". And add citations to 2 or 3 of the key papers here.

12) Page 2 -- line 11 -- change "the climate response to stratospheric aerosol injection" to "the climate response to a geoengineering-enhanced stratospheric aerosol layer" or similar. It's the eventual enhancement to the stratospheric aerosol layer that causes the forcing, not the injection. With a residence time of months to years, there is quite some difference between the nature of any injection and the resulting forcing that the climate then responds to.

13) Page 2 -- line 13 -- change "alter the climate" to "cool the surface climate and warm the stratosphere".

14) Page 2 -- line 19 -- change "Studies of SSG" to "Studies of SRM" -- and change other instances of "SSG" within the paper instead to "SRM"

15) Page 2 -- line 20 -- change "unfavorable size distributions" to "shorter residence time in the stratosphere (larger particles)".

16) Page 2 -- line 25 -- As I explained in my general comments, "These limitations might be addressed.." is not scientific language, and should be focused on which of the 5 limitations stated, emitted "various solid particles" the suggested solid particles is intended to address.

17) Page 2 -- line 25 -- The phrase "altering the size distribution" does not adequately communicate the complexity of the microphysical and dynamical processes that combine to effect the stratospheric aerosol layer's adjustment to a geoengineering source of aerosol particles. Whilst I understand that a strategy can be designed for engineered particles to aim to achieve a given desired size within the subsequent months to years of their circulation within the stratosphere, it is over-simplifying this to refer to "altering the size distribution".

It is of course certainly possible to alter the size distribution of the emitted particles, but any control within the response of the stratospheric aerosol layer in the subsequent months is too uncertain to be referred to simply as "altering the size distribution".

Please change "alternatively some of the limits may be addressed by altering the size distribution" to "with engineering strategies potentially able to achieve a more prolonged aerosol particle residence time in the stratosphere". Or if the authors mean the radiative efficacy, please phrase this more explicitly to be clear of the size effect intended.

18) Page 2 -- line 28 -- The sentence beginning "Efficacy is decreased" needs to be re-written for at least two reasons. Firstly, this is the first time the word "Efficacy" has been introduced, and it's not clear where this is scattering efficiency or efficacy in terms of residence time. The remainder of the sentence suggests it is mainly the latter -- so rather than "Efficacy is decreased...", suggest instead to say "Aerosol particle residence time in the stratosphere reduces...".

19) Page 3 -- line 15 -- "Our aerosol size distribution" is unscientific language. Please change to "The constant size distribution used by the models in the co-ordinated experiment..."

References
* * *
Dyer, A. J. and Hicks, B. B.
"Global spread of volcanic dust from the Bali eruption of 1963"
Q. J. Roy. Meteorol. Soc., vol. 94, pp. 545-554, 1968.

Dyer, A. J.
"The effects of volcanic eruptions on global turbidity, and an attempt to detect long-term trends due to man",
Q. J. Roy. Meteorol. Soc., vol. 100, pp. 563-571, 1974.

Grant, W. B., Browell, E. V., Long, C. S., Stowe, L. L., Grainger, R. G. and Lambert, A.:
"Use of volcanic aerosols to study the tropical stratospheric reservoir"
J. Geophys. Res., vol. 101, no. D2, pp. 3973-3988, 1996.

Pinto, J. P., Turco, R. P. and Toon, O. B.:
"Self-limiting physical and chemical effects in volcanic eruption clouds"
J. Geophys. Res., vol. 94, no. D8, pp. 11,165-11,174, 1989.

---

## Author Comment (AC1)

**Response to Reviewer 1**

We thank the reviewer for careful reading of our manuscript and constructive comments. Our responses to specific minor comments are given below in italics following each comment. Considering the issues raised by the reviewer has allowed us to improve our manuscript by clarifying these issues in the text. We are grateful for the reviewer's time and thoughtfulness.

Review for Weisenstein et al.,

"A model intercomparison of stratospheric solar geoengineering by accumulation mode sulphate aerosol"

Submitted ACPD

Here authors analyse CCM output from a dedicated GOIP solar engineering experiment "AM-H2SO4". This experiments is designed to inject geoengineering Sulphur (S) in the stratosphere in terms of particles (SO3 or H2SO4), so that stratospheric aerosol particles would grow mainly in accumulation mode, thereby negating effects of faster particle growth (and associated particle sedimentation). Analysis in this manuscript suggest that only three CCMs (WACCM, ECHAM5-HAM and SOCOL-AER) managed to complete these simulations. Basic idea behind these simulations is to differentiate model response to the SO2 vs particle injection under different (5 vs 25) Tg S injection magnitude scenarios. Authors find that all three models show increased radiative efficacy (in terms of radiative forcing) when Sulphur is injected in "AM-H2SO4" mode compared to gas phase injection. Also sensitivity simulations with different injection patterns (two points at 30° N and 30° S vs injection in a belt along the equator between 30° S and 30° N) find opposite response.

Overall this is well written manuscripts and fits well within ACP scope. Hence, I will like to recommend this manuscript for the publication with minor corrections.

Minor Comments:

1. Page 3: Line 28: Does that mean ECHAM has identical ozone loss in all the simulations?

   *Echam uses the same prescribed ozone and OH fields for the reference and geoengineering simulations. Thus there is no ozone loss due to geoengineering in the simulations. We have made this explicit in the text.*

2. Line 6 Line 18: I am really surprised that you use only 2 year spin up period. If you plot global burden, you would see steady increase in burden before curve

flattens, depending on dry and wet deposition schemes. Unless you have meteoric smoke particles transporting or mopping S-containing species downwards and there is lack of particle evaporation (temperature increase due to ozone increase), gas phase tracers (e.g. SO2, H2SO4) would show steady transport upwards. Overall tracers should reach to equilibrium state near model top after 3 to 4 years as they transport downward in the polar vortex. I think that is why WACCM (page 10 line 8) shows increasing residence with increase in injection amount. For e.g. Dhomse et al., 2013 (Figure 3) equilibrium for meteoric smoke particles is about 10 years. I suspect it should be at least 5 years for these simulations.

*While we expect that aerosol concentrations near the top of the sulfate layer and at high latitudes might still be evolving after 2 years, this paper analyzes only globally averaged quantities which we find to be fairly stable with time after 2 years for most scenarios. See enclosed plots of time evolution of global burden for the CESM2 and MAECHAM5-HAM models. Aerosol global burdens are still rising after 4 years with AM-H2SO4 injections of 25 Tg/yr but are stable after 2 years for the other scenarios. The SOCOL model simulations actually used a 5 year snmi-up period and then the following 8 years are averaged, so those results should not have an issue with spin-up length. For the CESM2 and MAECHAM5-HAM models, global aerosol burdens using averages of the last 5 or 6 years of the 10 year simulations are greater than for the 8 year averages by only 2-3% with AM-H2SO4 injections of 25 Tg/yr. We will acknowledge in the paper that the spin-up period was too short for these scenarios but has minimal effect on any of the quantities presented or conclusions drawn.*

*None of these simulations contains meteoritic smoke particles. The residence times of aerosol shown in Figure 3 of the paper are derived from burdens and injection rates (not diagnosed removal rates), with injection rates constant in time. Thus correcting for a too-short spin-up time in the AM-H2SO4 25 Tg/yr scenarios would increase the residence time for the 25 Tg(S)/yr injections in our plot and does not explain why CESM2/WACCM shows increasing residence with an increase in injection amount.*

[Figure]

*Figure R1:  Time evolution of global aerosol burden in the CESM2 simulations.*

[Figure]

*Figure R2:  Time evolution of global aerosol burden in the MAECHAM5-HAM simulations.*

3.  Page 6: Line 19: What is baseline or reference simulation? Do you mean from respective SSP8.5 simulation? Is it from a single ensemble member or from ensemble mean?

*The baseline scenario is described in the preceding paragraphs with SSP8.5 2040 GHGs and ODSs and SSTs for the 1988-2007 period. It is from a single ensemble member and we use an 8-year average from each model in our analysis.*

4.  Page 8 : line 1: Are you sure about only 10%? One needs to have very fast wet deposition. I think you should provide a line plot showing time variation in global burden.

    *The tropospheric burden is found to be 10% or less of the global aerosol burden increase in the SOCOL-AER model, even for the 25 Tg(S)/yr cases. Tropospheric burdens are not available for the other models.*

    *A line plot of global burden time variation is provided for the reviewre (figures R1 and R2 herer, but as we don't have saved data of the 5-year spin-up period for SOCOL-AER, we do not include these plots in the paper.*

5.  Page 9: line 1: it should be other way round : weaker stratosphere troposphere exchange in the SH hence more aerosol accumulate in SH mid-lats.

    *This referred to the lack of mixing between mid latitude and polar air in the southern hemisphere and hence the aerosol burden contrast in those two regions, which is not seen in the northern hemisphere with its weaker polar vortex and more efficient mixing to the pole. We modified the text to make this explicit.*

6.  Page 11: Figure 4: Does slope remain constant if you use only last 5 year data (5 year spin up).

    *Yes, using the last 5 years produces minimal impact of the figure.*

7.  Page 12: line 9 : Any idea why ECHAM shows much weaker sensitivity.

    *We are not able to diagnose the precise reason or reasons for the weaker sensitivity in the ECHAM model. Previous comparisons between ECHAM and CESM pointed to weaker tropical upwelling in ECHAM than CESM. The models also differ in the details of their aerosol formulations and their chemistry.*

8.  Page 18 : line 6 : Edit : 30°S-30°N

    *We have corrected this.*

9.  Page 21: line 19: Are you sure it is minor. In Dhomse et al (2015), it is about 3%. With significant Cly decrease, future ozone losses would be largely controlled by NOy chemistry (e.g. Ravishankara et al.,2009), I would expect up to 5% ozone increase in the tropical middle stratosphere.

*We do discuss ozone control by NOy chemistry and have added the reference to Ravishankara et al. (2009) and additional discussion regarding future ClOx and NOx changes. Ozone increases due to SO2 and AM-H2SO4 injections in these models are found in the 10-50 hPa region and near the tropical tropopause in the CESM model.*

References:

Dhomse, S.S., Saunders, R.W., Tian, W., Chipperfield, M.P. and Plane, J.M.C., 2013. Plutoniumâ©❺238 observations as a test of modeled transport and surface deposition of meteoric smoke particles. Geophysical Research Letters, 40(16), pp.4454-4458.

Dhomse, S. S., M. P. Chipperfield, W. Feng, R. Hossaini, G. W. Mann, and M. L. Santee (2015), Revisiting the hemispheric asymmetry in midlatitude ozone changes following the Mount Pinatubo eruption: A 3-D model study, Geophys. Res. Lett., 42, 3038–3047, doi:10.1002/ 2015GL063052.

Ravishankara, A.R., Daniel, J.S. and Portmann, R.W., 2009. Nitrous oxide (N2O): the dominant ozone-depleting substance emitted in the 21st century. Science, 326(5949), pp.123-125.

[Figure]
Reply
**Citation**: https://doi.org/10.5194/acp-2021-569-RC1

---

## Author Comment (AC2)

**Response to Reviewer 2**

We thank the reviewer for careful reading of our manuscript and constructive comments. Our responses to general and specific comments are given below in italics and indented following each comment. Considering the issues raised by the reviewer has allowed us to improve our manuscript with more precise wording. We are grateful for the reviewer's time and thoughtfulness.

Review of manuscript "A Model Intercomparison of Stratospheric Solar Geoengineering by Accumulation-Mode Sulfate Aerosols" by Debra Weisenstein et al.

This manuscript presents results from a model intercomparison comparing interactive stratospheric aerosol simulations within co-ordinated multi-model experiments to explore the global dispersion and radiative forcing that would result from a continuous source of sulphur dioxide or accumulation mode sulphate aerosol particles with two different emissions scenarios: none emitting only at 30N and 30S, the other as a constant source between 30S and 30N.

The intercomparison compares results from 3 different interactive stratospheric aerosol models (WACCM-MAM3, MAECHAM5-HAM and SOCOL-AER), and represents a potentially very interesting contribution to understand the predictions from the models, each having differing sophistication in their aerosol modules, and in the vertical and horizontal resolution of the GCM's advection.

Whilst the results are interesting, and certainly will be publishable in a revised form, the aim and design of the model experiments are surprisingly poorly described, and the Introduction and interpretation need to include some discussion also of the tropical stratospheric reservoir, in relation to the differences between the two scenarios.

In several places the manuscript has unscientific language and vague statements that need to be changed to terms more appropriate to a journal article. For example "may produce overly large aerosols" (page 1, line 12) and "unfavorable aerosol size distributions" (page 2, line 20) and "Our aerosol size distribution" (page 3, line 14) are clearly subjective terms that need to be better phrased to communicate the issues involved.

> *We have modified the writing to be more scientifically precise and thank the reviewer for pointing this out.*

There are also a few places where the wording is poor, for example "These limits might be addressed" (page 2, line 25), "some of these limits may be addressed by altering the size distribution of sulfate aerosol" (page 2, lines 26-27). The authors are clearly aware that these issues are at the heart of the science to understand the

efficacy and risk of a hypothesised large-scale injection of precursor gas of idealised particle for solar radiation management. Referring to "altering the size distribution of the sulfate aerosol" grossly simplifies the complex interplay of processes involved -- and the wording needs to communicate consistently with an awareness of these issues.

I also find it very surprising that, in this initial version of the manuscript, the authors have not adequately explained the rationale for the very interesting model experiments they are presenting results from. The two "injection scenarios" presented in the paper: 1) emitting continuously at two sites at 30N and 30S, and 2) emitting continuously throughout all latitudes between 30N and 30S, not surprisingly cause very different enhancements to the stratospheric aerosol layer (as is clearly seen in Figure 2). The 30N and 30S two-site scenario causes the stratospheric aerosol layer enhancement to be almost exclusively in mid- and high-latitudes, with only a very minor elevation in stratospheric aerosol optical depth in the tropics. The main reason for this is of course that the two-site injection scenario emits SO2/particles entirely outside the tropical stratospheric reservoir (between 20S and 20N). It is well established (e.g. Dyer, 1974) that the residence time for volcanic aerosol clouds formed in the tropical stratosphere are much longer (around 2 years) than for eruptions forming stratospheric aerosol clouds in the mid-latitudes. The reason is the continuing tropical upwelling and the transport barrier at the edge of the tropical pipe, and analysis of satellite measurements in 1991-1992 show the effect for example on the dispersion of the Pinatubo aerosol cloud (see Grant et al., 1996 for example). There needs to at least be sentence briefly mentioning the tropical stratospheric reservoir in the Introduction, and some discussion of the seasonal cycle of the Brewer-Dobson circulation (e.g. as set out originally by Dyer et al., 1968 for the Agung aerosol cloud).

Just to be clear, my review is not saying these results are not interesting, the results are indeed very interesting --- and this is a laudible effort to have a set of experiments to better understand any differences in predictions with the models -- but there needs to be a much clearer explanation of the rationale for why these scenarios were chosen.

It's implicit in the text that the 30N and 30S case might represent a limited 2-site injection strategy, but it needs to be made clear that the two scenarios are not really comparable, and that our understanding of stratospheric circulation would clearly mean that the 2-site 30N and 30S injection scenario would give a mid-latitude focussed stratospheric aerosol forcing, whereas the 30S-to-30N area-source scenario is presumably designed to give a more evenly-spread stratospheric aerosol layer enhancement, with then substantial radiative forcing also in the tropics.

> *We have modified the manuscript to more fully describe the rationale for the regional and 2point injections in both the abstract and scenario descriptions.*

I'm also recommending the authors consider changing the title, because the term "Solar Geoengineering by Accumulation-Mode Sulfate Aerosols" is not consistent with what the authors state that particular model experiment is representing.  Firstly, the models ran separate simulations with continuous emission of SO2, in addition to the experiment with the particle source, so the experiments are to explore also the injection of SO2, in addition to direct injection of particles. So the title should either state that both are carried out, or else just give a more general summary-term there.

> *We have modified the title to read:  "An aerosol-climate model intercomparison of stratospheric solar geoengineering by injection of SO2 or accumulation-mode sulfate aerosols".*

Secondly, the manuscript states (page 3, lines 14-17) that the particle-injection experiment is designed to represent particle sizes that would occur at the grid-scale of the large scale models following injection of SO3 or H2SO4 from high altitude aircraft, a localised plume subsequently generating a source of accumulation mode particles at the grid-scale of the GCMs.  The text states "Our aerosol size distribution is consistent with Pierce et al. (2010) and Benduhn et al. (2016) who modelled plume microphysics and found that injection rate could be adjusted to produce sulfate aerosol size distrihbutions in the 0.1-0.15 micron radius size range."

I'd say first that I'm not sure either of those two first authors would argue that one can simply adjust the injection rate to produce the desired size distribution. I would expect that both would explain that there would be a substantial variability in the size distribution generated as the plume subsequently entrains into, and becomes mixed with the surrouding ambient air.  So I'd argue that the text "could be adjusted to produce" is not really adequately representing the eventual variability in sizes that would result there.  That said, I accept that a large diversity range for particle size is given (0.1 to 0.15 microns).  It's again a case of the wording not adequately communicating the issues involved.

> *We have modified the wording here:  "Our aerosol size distribution is consistent with Pierce et al. (2010) and Benduhn et al. (2016) who modelled plume microphysics and found that sulfate aerosol size distributions in the 0.1-0.15 μm radius size range could potentially be produced.  Detailed modelling of this complex process for a full range of stratospheric physical, chemical, and microphysical conditions awaits further studies."*

In my specific comments below, I'm recommending the authors consider using the terminology "sub-grid-scale sulphate emission" rather than "accumulation-mode particle emission", or as an alterntive they could actually explain the rationale of the experiment is to represent a proxy for an idealised particle source, deliberately designed to produce particles at a particular desired size.

In light of the likely large variability in particle sizes that continued gaseous emission of SO3 or H2SO4 would cause, to me it is actually this engineered particle-emission scenario that these controlled size-distribution experiments are representing.

*We now acknowledge this: "These GCMs can effectively simulate changes in global aerosol burden, radiative forcing, ozone, and stratospheric temperature and circulation. As input, they would take the particle size distributions from aircraft plume model studies but could represent any hypothetical input of particles. The input size distribution is simplified here by using a single mode radius and mode width at all grid points and times."*

The other similar terminology issue I identify in my specific comments, is that the authors seem to use both the acronym "SRM" for solar radiation management, and also use the acronym "SSG" for stratospheric solar geoengineering. In my view, the paper needs to be consistent in either using SRM or SSG, but not both. My recommendation would be to use the acronym SRM, since the acronym SSG is often used for "scientific steering group", and SRM is also (in my mind) the more established term.

*We have replaced "SSG" with "SRM" throughout the paper for consistency.*

I'm suggesting the title should also be clear these are interactive stratospheric aerosol simulations being intercompared, with my suggestion being to change the title from "A model intercomparison of stratospheric solar geoengineering by accumulation-mode sulfate aerosols" instead to something like: "A co-ordinated intercomparison of interactive stratospheric aerosol model experiments for hypothesised scenarios of solar radiation management by sulfate aerosols"

We have modified the title to read: "An aerosol-climate model intercomparison of stratospheric solar geoengineering by injection of SO2 or accumulation-mode sulfate aerosols".

I provide below a list of specific comments I am asking the authors to address, and with these comments requesting a change in the tone of the narrative of the manuscript, my review then finds major revisions are needed. The authors may find it relatively easy however to make these changes -- with the Figures, and much of the results section is in good shape, requiring only minor revisions.

Specific comments:
* * *
1) Page 1 -- lines 1-2 -- Further to the comments above, I strongly recommend the authors consider using a different term than "solar geoengineering by accumulation-mode sulfate aerosols".

The optical depth from the stratospheric aerosol layer mainly comes from sulfate aerosols in the accumulation mode size range, and the forcing from any geoengineered enhancement to the stratospheric aerosol layer would be caused by particles in the accumulation mode part of the size spectrum. So using the precursor term "accumulation-mode" prior to the "sulfate aerosols" is not really a useful descriptor of the effect.

I realise that one of the co-ordinated multi-model experiments involves each model adding a continued source of sulfate aerosol particles at a particular constant size distribution (in the accumulation mode size range) but that term is then referring to some specifics of the design of the model experiment.

Remember that the residence time of particles in the stratosphere is months to years and the resulting size distribution from a continued emissions is rather a response to that source of particles, and the microphysical and dynamical processes likely mean the resulting size distribution may differ substantially from that within a localized primary emission. That does not necessarily rule out devising a source of particles engineered to achieve a particular resulting size distribution.  But a terminology referring simply to "solar geoengineering by accumulation-mode sulfate aerosols" could lead to some readers inferring too simplified a relationship between the size distribution at particle emission and the evolving size distribution of the resulting dispersed aerosol cloud.

The authors refer to Pierce et al. (2010) and whilst the 2D-AER interactive stratospheric aerosol simulations give a reasonable assessment for the progression of the geoengineered aerosol cloud, the dilution of the initial plume and its subsequent evolution of the size distribution of the dispersed aerosol within the stratospheric dynamics of a higher resolution 3D GCM may well have given differing result.

As I say, I am not at all underplaying the value of these model experiments, which could well help to shed light on some of these issues, but I strongly advise the authors use a different terminology for the mechanism the model experiments are investigating.

Within the article, the authors need to be clearer whether these experiments really are representing a scenario of injected H2SO4 vapour. With the resulting plume rapidly nucleating particles to form a source of new particles that progress to be large enough to scatter incoming solar radiation.

The current model experiments do not really represent that situation, because there would certainly be greater variability in the size distribution in that case of H2SO4 vapour emission.  Rather I would argue these experiments mimic a situation where particles are emitted with a controlled size distribution, the particles deliberately engineered to achieve a certain subsequent response within the stratospheric aerosol layer.

My recommendation in this first comment is to change to a more general title something like:

"A co-ordinated intercomparison of interactive stratospheric aerosol model experiments for hypothesised scenarios of solar radiation management by sulfate aerosols"

That's partly because the experiments are not restricted to only assess an emitted source of particles, they also assess the models' response to emitted SO2. In light also of the potentially large variations in particle size distribution that would result, to simply tag the approach as "Accumulation mode particle geoengineering" is not appropriate (in my opinion).

As per the subsequent specific comments, within the article, I can understand there is a benefit to referring to the effect from the strategy (in that it is specifically introducing accumulation mode particles into the models), but still I'm recommending the authors use a different terminology than "AM-H2SO4 geoengineering".

Global aerosol microphysics modellers may tend to use the term "sub-grid scale particle formation" or "primary sulphate emission" for this approach, with the former being much preferred to the latter by experts. And in my comments then I advise to use the term "sub-grid scale particle formation model experiments" or similar as the alternative term.

> *We prefer to retain the term "AM-H2SO4" as it is less wordy than alternatives. However, we add more description of these scenarios to avoid oversimplification and misleading readers.*

2) Page 1 -- Abstract, line 11 -- Suggest to insert "tended to focus on" rather than simply "focussed on", and rather than the somewhat vague term "Analyses", be clear you're referring to interactive stratospheric aerosol model analyses". In fact probably better to use "studies" rather than "analyses".

> *Changed first part of sentence to read "Model studies of stratospheric solar geoengineering have tended to focus on sulfate aerosol enhancement..."*

3) Page 1 -- Abstract, lines 11-12 -- the 2nd half of this 1st sentence then refers to climate models (whereas I think the first half refers to interactive stratospheric aerosol models). I think I understand what the authors mean when they say the climate model experiments "have assumed injection of SO2", but that could confuse some readers, because the majority of climate models do not tend to use their interactive aerosol modules for stratospheric aerosol, and therefore do not tend to represent injection of SO2 at all.

I think what the authors mean is that the model experiments tend to be designed to represent a scenario of imposing radiative effects consistent with best estimates of what could be expected from continued injection of SO2.

I suggest to change "have assumed injection of SO2" to "are based on scenarios aimed to represent the effects from continued SO2 injection". Or similar.

> *Changed this part of sentence to: "and almost all such climate model experiments are base on scenarios which assume injection of SO$_2$ for this purpose."*

4) Page 1 -- Abstract, line 12 -- As per my general comments above, "may produce overly large aerosols" is obviously unscientific language. Also, that particles grow larger with increased SO2 is a scientific fact, with then use of the word "Yet" not good grammar.  It's an important point the authors are making, but this should be stated in an objective way, whereas the precursor word "Yet" suggests the authors consider it somehow unfortunate or undesirable.

Suggest "It is well established (e.g. Pinto et al., 1989) that greater emission of SO2 leads to larger sulphate aerosol particles, with shorter residence time in the stratosphere."

> *Suggested wording adopted and used in the introduction, to avoid references in the abstract.*

5) Page 1 -- Abstract, line 13 -- I think changing "new" to "additional" changes to a more accurate representation, to ensure authors do not mistakenly infer that particles form immediately at accumulation mode sizes (but rather grow from an initially smaller Aitken mode sizes) in this scenario of aircraft injection of SO3 or H2SO4.

This is an example of where I think the simplified term "geoengineering by accumulation mode sulphate" might only increase the probability of an incorrect inference in that respect.  I therefore strongly suggest the authors delete "AM-H2SO4", as the acronym similarly will tend to embed an increased likelihood of that over-simplified perception of the progression of the microphysical and dynamical processes involved.

The term "nudged" is also not appropriate in this context, tending to over-simplify the response of the stratospheric aerosol layer.

I'd suggest to re-word to "Some studies have explored whether a stream of very small particles can be generated by injecting H2SO4 vapour rather than SO2, potentially then leading to longer-lived aerosol particles for a given sulphur injection rate."

Introducing a specific delivery mechanism seems un-necessary, and my suggested re-wording then also keeps the point more general than that specific situation of aircraft injection.

> *We have modified the abstract to read:  "Injection of $SO_3$ or $H_2SO_4$ from an aircraft in stratospheric flight is expected to produce additional accumulation-mode particles (AM-$H_2SO_4$) after microphysical processing within an expanding plume, and such injection may allow the resulting stratospheric sulfate aerosol layer to more effectively scatter solar radiation."*

6) Page 1 -- Abstract, line 15 -- For the reasons given earlier, please change the terminology "AM-H2SO4 injection" to refer to the specifics of the model experiments rather than an apparently more general "type of geoengineering".  As

explained in my comments above, I'm suggesting to use the term "sub-grid scale source of particles" as the descriptor, referring then to the specifics of the model experimetns, with also an acronym then not required in this case.

> *We prefer to retain AM-H2SO4 but have clarified the term and it's implicit sub-grid processing:* "*Injection of $SO_3$ or $H_2SO_4$ from an aircraft in stratospheric flight is expected to produce additional accumulation-mode particles ($AM\text{-}H_2SO_4$) after microphysical processing within an expanding plume, and such injection may allow the resulting stratospheric sulfate aerosol size distribution to more effectively scatter solar radiation. . We report the first multi-model intercomparison to evaluate the effects of such an approach, which we label $AM\text{-}H_2SO_4$ injection based on the size distribution input to global-scale models after implicit sub-grid processing.*"

I suggest then to change this sentence to instead say "Whereas GeoMIP has included experiments to intercompare SO2 injection scenarios, the results here are the first multi-model intercomparison of the effects from a sub-grid scale source of sulphate aerosol. Or something like this.

With the subsequent sentence referring to GeoMIP, suggest to reserve the statement of "first" for after the sentence referring to GeoMIP. The scope of that sentence can be made more general by changing "We compare three models" to "A co-ordinated multi-model experiment designed to represent this SO3- or H2SO4-driven geoengineering scenario was carried out with 3 interactive stratospheric aerosol models:". The word "coordinated" can then be deleted later in the sentence.

> "*We report the first multi-model intercomparison to evaluate and compare the effects of such an approach, which we label $AM\text{-}H_2SO_4$ injection based on the size distribution input to global-scale models after implicit sub-grid processing.* *A co-ordinated multi-model experiment designed to represent this $SO_3$- or $H_2SO_4$-driven geoengineering scenario was carried out with 3 interactive stratospheric aerosol-climate models...*"

7) Page 1 -- Abstract, line 24 -- Further to my general comments above, the term "sensitivity to injection pattern" is not an adequate description of the two experiments, the two-site experiment resulting in a midlatitude-focused forcing with little enhancement to the tropical stratospheric reservoir. The word "sensitivity" suggests a slight change whereas these two alternative representations of the geoengineering enhancement are much more substantially different. Better to actually crystallise in the reader's mind what the two alternative scenarios represent -- a mid-latitude-focussed forcing (presumably designed to avoid perturbing climate-sensitive regions in the tropics?) and an evenly distributed injection rate across the tropics and mid-latitudes.

Suggest then to change the sentence beginning "We explore the sensitivity to injection pattern" to "Simulations with two scenarios were designed to compare a two-site injection focused to force only the mid-latitudes, with a more evenly distributed geoengineering forcing, each run with both SO2 and sub-grid particle experiments." or something like this.

The "and find opposite impacts" is explaining the results, and should be explained in a separate sentence, changing "and find opposite" to "We find opposite" or similar.

> *We have revised this part of the abstract to read: "We use two different injection patterns, one designed to force mainly the midlatitudes by injecting at 30° N and 30° S and the other designed to force more uniformly by injecting in a belt along the equator between 30° S and 30° N. By forcing each case with both $SO_2$ and AM-$H_2SO_4$, we find opposite impacts on radiative efficacy for the two injection patterns, suggesting that prior model results for concentrated injection of $SO_2$ may be strongly dependent on model resolution."*

8) Page 2 -- Introduction, line 2 -- insert "long-wave" before "radiative forcing" and change "from the rise in CO2" to "from increased CO2 concentrations". The reason here is to keep in the reader's mind that emissions are not necessarily the same as concentrations.

> *Suggested language adopted.*

9) Page 2 -- Introduction, line 3 -- "Despite the complexity" -- it's not corect to say "Despite" here and I'd argue it's more "because of the complexity" that these models are needing to be used to try to predict how the overall system responds given the complex interactions and feedbacks.

Suggest to change "Despite the" to "In light of the".

> *Suggested language adopted.*

10) Page 2 -- Introduction, line 4 -- "solar radiation management (SRM) is being studied". It's not really the solar radiation management itself that is being studied -- it's the effects from hypothesized solar radiation management (whether that be the responses of the stratospheric aerosol and ozone layers or the surface response to climate and the hydrological cycle).

Suggest to insert "the effects from hypothesized" after "carried risks".

> *Suggested language adopted.*

11) Page 2 -- line 5 -- since the word "climate" is used later in the sentence (and with the change above also earlier in the sentence) change "climate models" to "earth system models". And add citations to 2 or 3 of the key papers here.

> *Suggested language adopted. Full sentence now reads: "In light of the complexity of the climate system and the inherent risks of climate manipulation, the effects of hypothesized solar radiation modification (SRM) are being studied with earth system models to examine the potential benefits and possible adverse effects (e.g. Aquila et al., 2014; Richter et al., 2017; Tilmes e al., 2017) while simultaneously improving our knowledge of climate interactions and feedback processes."*

12) Page 2 -- line 11 -- change "the climate response to stratospheric aerosol injection" to "the climate response to a geoengineering-enhanced stratospheric aerosol layer" or similar.

It's the eventual enhancement to the stratospheric aerosol layer that causes the forcing, not the injection.  With a residence time of months to years, there is quite some difference between the nature of any injection and the resulting forcing that the climate then responds to.

*Suggested language adopted.*

13) Page 2 -- line 13 -- change "alter the climate" to "cool the surface climate and warm the stratosphere"

*Suggested language adopted.*

14) Page 2 -- line 19 -- change "Studies of SSG" to "Studies of SRM" -- and change other instances of "SSG" within the paper instead to "SRM"

*Suggestion adopted.*

15) Page 2 -- line 20 -- change "unfavorable size distributions" to "shorter residence time in the stratosphere (larger particles)".

*Changed "unfavorable size distributions" to "larger particles (less efficient shortwave scattering) and shortened aerosol residence time"*

"16) Page 2 -- line 25 -- As I explained in my general comments, "These limitations might be addressed.." is not scientific language, and should be focused on which of the 5 limitations stated, emitted "various solid particles" the suggested solid particles is intended to address.

*Changed this to "Limitations (3) and (4) might be addressed through use of various solid aerosol particles for SRM"*

17) Page 2 -- line 25 -- The phrase "altering the size distribution" does not adequately communicate the complexity of the microphysical and dynamical processes that combine to effect the stratospheric aerosol layer's adjustment to a geoengineering source of aerosol particles.  Whilst I understand that a strategy can be designed for engineered particles to aim to achieve a given desired size within the subsequent months to years of their circulation within the stratosphere, it is over-simplifying this to refer to "altering the size distribution".  It is of course certainly possible to alter the size distribution of the emitted particles, but any control within the response of the stratospheric aerosol layer in the subsequent months is too uncertain to be referred to simply as "altering the size distribution".

Please change "alternatively some of the limits may be addressed by altering the size distribution" to "with engineering strategies potentially able to achieve a more prolonged aerosol particle residence time in the stratosphere".  Or if the authors mean the radiative efficacy, please phrase this more explicitly to be clear of the size effect intended.

> *Changed to "alternatively limitation (1) may be addressed with geoengineering strategies designed to achieve a sulfate aerosol layer with a size distribution that optimizes shortwave scattering"*

18) Page 2 -- line 28 -- The sentence beginning "Efficacy is decreased" needs to be re-written for at least two reasons. Firstly, this is the first time the word "Efficacy" has been introduced, and it's not clear where this is scattering efficiency or efficacy in terms of residence time.  The remainder of the sentence suggests it is mainly the latter -- so rather than "Efficacy is decreased...", suggest instead to say "Aerosol particle residence time in the stratosphere reduces...".

> *Suggestion adopted.*

19) Page 3 -- line 15 -- "Our aerosol size distribution" is unscientific language. Please change to "The constant size distribution used by the models in the co-ordinated experiment..."

> *Suggestion adopted.*

References
* * *
Dyer, A. J. and Hicks, B. B. "Global spread of volcanic dust from the Bali eruption of 1963" Q. J. Roy. Meteorol. Soc., vol. 94, pp. 545-554, 1968.

Dyer, A. J. "The effects of volcanic eruptions on global turbidity, and an attempt to detect long-term trends due to man", Q. J. Roy. Meteorol. Soc., vol. 100, pp. 563-571, 1974.

Grant, W. B., Browell, E. V., Long, C. S., Stowe, L. L., Grainger, R. G. and Lambert, A.: "Use of volcanic aerosols to study the tropical stratospheric reservoir" J. Geophys. Res., vol. 101, no. D2, pp. 3973-3988, 1996.

Pinto, J. P., Turco, R. P. and Toon, O. B.: "Self-limiting physical and chemical effects in volcanic eruption clouds" J. Geophys. Res., vol. 94, no. D8, pp. 11,165-11,174, 1989.

> *Three of these four references have been included in the revised manuscript.*

[Figure]

**Citation**: https://doi.org/10.5194/acp-2021-569-RC2

---

## Author Comment (AC3)

**Response to Reviewer 3**

We thank the reviewer for careful reading of our manuscript and constructive comments. Our responses to the reviewer's general and specific comments are given below in italics following each comment. Considering the issues raised by the reviewer has allowed us to improve our manuscript by clarifying these issues in the text. We are grateful for the reviewer's time and thoughtfulness.

- This study investigates the implications of using $SO_3$ or $H_2SO_4$ instead of $SO_2$ in deliberate emissions in the stratosphere in order to modify Earth's climate. Using $SO_3$ or $H_2SO_4$ would produce smaller particles (accumulation mode – AM-$H_2SO_4$) which are more radiative effective than those formed from emissions of $SO_2$. The effects of geoengineering with AM-$H_2SO_4$ is investigated using three global climate models. The effects on the stratospheric size distribution, aerosol load, temperature, water vapour and ozone as well as the radiative effects are investigated. All models show that there is increased radiative efficiency using AM-$H_2SO_4$ but there are large intermodel differces.

  The study is well performed and many different aspects of using AM-$H_2SO_4$ instead of $SO_2$ is investigated. This type of investigation using three models in one study has not been performed before. The three models used in the study have different strength and weaknesses in their representation of the stratosphere which gives relevant information of the uncertainties in the modelling geoengineering in the stratosphere with AM-$H_2SO_4$ and $SO_2$. The paper is well written in general and has a clear structure. The paper is well within the scope of ACP and I recommend publication after the following comments has been addressed.

  **General comments:**

  It would be interesting to include a short discussion on the feasibility of using $SO_3$ or $H_2SO_4$ instead of $SO_2$ and whether one of the options is more technically challenging than the other one.

  *We have added a sentence regarding the technical and engineering challenge of using SO3 or H2SO4 and included two references,* Smith et al., 2018 and Janssens et al., 2020.

  **Specific comments:**

  Page 6, line 27: Why were the emissions released at different heights in the different models?

  *This was a function of the model's vertical grid resolution and necessary conversion from altitude to pressure level.*

  Page 6, line 30: I miss an explanation or motivation of the choice of the different injections and injections points. What was the scientific motive for choosing those emissions and emissions points? Which scientific questions could be answered with these?

  *We added more explanation of the injection patterns in the scenario descriptions in Section 2: "The regional injections are designed to utilize the Brewer-Dobson circulation to distribute emissions globally and maximize their residence time, as has been observed for volcanic aerosol clouds (Dyer, 1968; Grant et al., 1996). The 2point injections occur outside the tropical stratospheric reservoir (Grant et al., 1996; Tilmes er al., 2017) and are meant to concentrate geoengineering impacts at higher latitudes and to explore microphysical differences when injections are more concentrated spatially."*

Page 12, line 26: "main particle size distribution from an $R_g$". What is the main size distribution $R_g$? $R_g$ was defined as the mode radii value, but the main size distribution cannot have one mode radii value.

*Corrected to be the accumulation mode.*

Page 24, line 11-16. There is quite a lot of discussion here that has not been included previously in the manuscript. The section head should perhaps be changed from "summary and conclusion" to "summary and discussion."

*Adopting this suggestion.*

**Technical corrections:**

Page 9, line 7: It is a bit vauge to start the sentence with "This figure" no figure has been mentioned for several sentences.

*Replaced "This figure" with "Figure 2".*

Page 10*, line 10-14. This sentence is very long. Please divide it.*

*Done.*

Page 13, line 7: This sentence is awkward, please revise.

*Revised to read: "The size distributions respond differently to 2point rather than regional injections depending on whether $SO_2$ gas or $AM$-$H_2SO_4$ particulate is injected. These results suggest the way aerosol microphysics drives differences between $AM$-$H_2SO_4$ and $SO_2$ injection scenarios (see Table 3)."*

Figure 11: The legend in this figure uses $SO_2$ and $H_2SO_4$ to denote the simulations rather than AM-$H_2SO_4$as in the rest of the manuscript. Please revise for consistency.

*Corrected.*

Reply

**Citation**: https://doi.org/10.5194/acp-2021-569-RC3

---

## Author Response (AR2)

**Report #2 by Reviewer #2**

We thank the review for another careful reading of our manuscript. His/her comments have helped us clarify several points and improve the manuscript. In two cases the reviewer misinterpreted our meaning and therefore we have clarified the text and stated our meaning more explicitly. In other cases we have adopted the reviewer's advise and appreciate the rewording suggestions. Our specific replies to reviewer comments 1 through 6 are listed below in italics and green text.

Submitted on 29 Dec 2021
Anonymous Referee #2

**Anonymous during peer-review: Yes** No

**Anonymous in acknowledgements of published article: Yes** No

**Recommendation to the editor**

| | |
|---|---|
| **1) Scientific significance**
Does the manuscript represent a substantial contribution to scientific progress within the scope of this journal (substantial new concepts, ideas, methods, or data)? | Outstanding **Excellent** Good Fair Low |
| **2) Scientific quality**
Are the scientific approach and applied methods valid? Are the results discussed in an appropriate and balanced way (consideration of related work, including appropriate references)? | Outstanding **Excellent** Good Fair Low |
| **3) Presentation quality**
Are the scientific results and conclusions presented in a clear, concise, and well structured way (number and quality of figures/tables, appropriate use of English language)? | Outstanding Excellent Good **Fair** Low |

For final publication, the manuscript should be

**accepted as is**

accepted subject to **technical corrections**

**accepted subject to minor revisions**

reconsidered after **major revisions**

**rejected**

**Were a revised manuscript to be sent for another round of reviews:**

**I would be willing to review the revised manuscript.**

I would not be willing to review the revised manuscript.

**Suggestions for revision or reasons for rejection** **(will be published if the paper is accepted for final publication)**

Review of "post-review revised" version of manuscript "A Model Intercomparison of Stratospheric Solar Geoengineering by Accumulation-Mode Sulfate Aerosols" by Debra Weisenstein et al.

I was one of the 3 reviewers who reviewed the submitted version of this manuscript in August 2021, the analysis presenting results from a model intercomparison comparing interactive stratospheric aerosol simulations within co-ordinated multi-model experiments to explore the global dispersion and radiative forcing that would result from a continuous source of sulphur dioxide or accumulation mode sulphate aerosol particles with two different emissions scenarios: one emitting only at 30N and 30S, the other as a constant source between 30S and 30N.

As I identified in my original review, the intercomparison (across 3 different models), represents a potentially very interesting contribution to understand the predictions from the models, each having differing sophistication in their aerosol modules, and in the vertical and horizontal resolution of the GCM's advection.

My original review explained the analysis would be publishable in a revised form, my review found the aim and design of the model experiments to be poorly described, with the Introduction and interpretation needing to include some discussion also of the tropical stratospheric reservoir, in relation to expected differences between the two scenarios.
I also noted several places where the manuscript had unscientific language or vague statement, and a few places where the wording was poor, or over-simplifying the changes to the stratospheric aerosol layer that would occur in this hypothesised large-scale injection of precursor gas or idealised particle for solar radiation management.

I made a list of 20 specific revisions that were needed, before the manuscript could potentially then proceed to publication, with also a request that the authors change the title, and querying the terminology "Solar Geoengineering by Accumulation-Mode Sulfate Aerosols", in being inconsistent with what the authors state that particular model experiment is representing.

With some of the comments relatively fundamental to the narrative of the manuscript, I found that major revisions were needed, but with the Figures much of the results section is in good shape, these were relatively minor revisions.

The authors have replied positively to each of the specific comments I made, with the instances of imprecise wording in the manuscript now remedied, with also most of the instances where particle size changes were over-simplified now also much improved.

One of my main comments was to advise the authors to change the term "geoengineering by accumulation-mode sulfate aerosols" to simply "geoengineering by sulfate aerosols.
This suggestion was on the basis that that terminology also somehow suggests a relative ease of achieving that particular desired size, or that the particles will remain at a particular "target size" in their months to years lifetime in the stratosphere.

As I explained in my review, the vertical and meridional variations in stratospheric aerosol particle size seen in the years after the Pinatubo eruption, and other major eruptions, and from my own experience from analysing a range of major eruption scenario experiments within interactive stratospheric aerosol models suggest the range of variation in particle size would likely be broader than many readers would infer from that "geoengineering by accumulation-mode sulfate aerosols" terminology. Somehow that terminology suggests, to me

at least, is that it is a relatively simple issue to achieve the required particle size.

In their reply to this comment, the authors explain that their preference to retain that same terminology -- and of course this is their manuscript -- and given the improvements in the wording the authors have made, despite my continued opposition to that term (it still communicated an over-simplified situation), it is OK for that terminology to remain in place within this particular article.

Although the manuscript is much improved, there are still some places where the wording requires minor changes, to then be fine to proceed to publication. These remaining few minor revisions are listed below, with 2 of the 6 required changes being more substantial than the others, and requiring some explanation here.

Both of these two remaining "substantive points" relate to the way the text in the revised manuscript refers to the specific type/class of models whose results are analysed in this multi-model analysis.

The first of the 2 is to ask the authors to change the name they have used for the type/class of models used in the analysis, specifically within the new title the authors have added in the revised manuscript, where the models are referred to as "aerosol-climate models".

Whilst I accept that the models can generally be referred to within that broad class of models that include the radiative effects of aerosol in their predictions of the earth's climate, the functionality being applied for this analysis, requires a particular capability for "interactive stratospheric aerosol". As the authors will be aware, there is currently ongoing a model intercomparison project/iniative "ISA-MIP" (Timmreck et al., 2018) which has designed model experiments to specifically inter-compare these interactive stratospheric aerosol models, including more background (volcanically quiescent) conditions, and experiments for major eruptions, and to hindcast predictions through the series of more moderate stratospheric-injecting eruptions that have occurred so far in the 21st century.

For that MIP, the terminology the community agreed for these models was "interactive stratospheric aerosol models" (the ISA within ISA-MIP) and whereas these models are certainly aerosol-climate models, the interactive stratospheric aerosol capability is key here in terms of being able to predict the onward variations in particle size, and associated residence time, from the initial "emission size distribution" the particles initially as they mixed into the ambient air around the aircraft.

And that 1st of the 6 changes is then requesting to change the title from "An aerosol-climate model intercomparison" instead to "An interactive stratospheric aerosol model intercomparison".

*We have modified the title as suggested, using the terminology adopted for ISA-MIP.*

It is notable to me that each of the 3 interactive stratospheric aerosol models compared here also having the particular sophistication to represent microphysical processes within their interactive predictions of the stratopheric aerosol layer's variations. And then I am advising here to also add the word "microphysics" before "intercomparison" -- this then being an "interactive stratospheric aerosol microphysics intercomparison. I leave it up to the authors however to choose either of those -- each an improvement on the much less specific "aerosol-climate model" terminology, which is too broad to adequately communicate the particular type of experiments the manuscript analyses results from.

The other of the 2 remaining substantive change also relates to the type of model, where I think the authors current description of the "modus operandi" of the models needs changing.

Specifically the wording:

"As input, they would take the particle size distribution from aircraft plume model studies
but could represent any hypothetical input of particles".

This reference to an "input size distribution" under-plays
the value of the interactive stratospheric aeorsol models, and could lead some readers to
mis-understand the aerosol-climate models to simply be enacting a "prescribed but globally varying"
size distribution from some other experiments with an interactive stratospheric aerosol model.
This is not the case, with the interactive models doing much more than simply representing a
particular "input size distribution".

Each of the 3 interactive stratospheric aerosol microphysics models used in the study predict how
the initially localised plume of geoengineering aerosol would progress to the "response" of a
global enhancement to the stratospheric aerosol layer, and how that would evolve in the months
of years of the continued injection/emission.

The global spatial variation of the particle size, and its temporal variability across different
seasons and years, with the internal variations in the stratospheric circulation and its dynamical
states that occur through the simulations. The relationship between a particular engineered
particle size at emission, and the eventual variation in size one would see at a global scale
(across the tropics, midlatitudes and high latitudes of each hemisphere) is far from certain.
In addition to the transport variations, the coagulation of the particles, and the subsequent
removal and vertical distribution from gravitational settling, will likely cause substantial
variations in the progressions in particle sizes that would occur as the continuing (or
intermittent) plume(s) are dispersed globally over months and years, would introduce subsequent
variations that differ from the initial "emission size distribution" introduced from the
emission location (e.g. aircraft or tethered pipe).

This issue communicates a bit more about the basis of my objection to my the
"geoengineering by accumulation-mode sulphate" terminology, in some readers inferring an
over-simplified situation in how the dispersed particles would "end up" within the "enhanced
state" of a geoengineered stratospheric aerosol layer.

Whilst I am content to concede that to the authors choice of terminology, when referring to
the models being used in the analysis, the text needs to better communicate what the interactive
stratospheric aerosol models actually represent. In particular the reference to an
"input size distribution", seems to be an over-simplfication.

*We realize the confusion between "input size" and global aerosol distribution.  Therefore we have
changed the wording to refer to a "mass flux of particles" or "source of particles at a specified rate".
This should clarify the difference between the input particle sizes and the resulting global aerosol
distribution.  On page 4, lines 3-4 now read:  "As input, the global models would take a mass flux of
particles with the size distributions generated by aircraft plume model studies or any hypothetical
source of particles at a specified rate."*

The other 4 changes are more minor, and self-explanatory, and I then list below the 6 minor revisions
which I'm advising are required before the manuscript can then proceed to publication in ACP.

Remaining minor revisions
* * *
1) Manuscript title: Page 1, line 1 -- please change "An aerosol-climate model intercomparison"
to either "An interactive stratospheric aerosol model intercomparison" or preferably (from my perspective)
"An interactive stratospheric aerosol microphysics model intercomparison". I note that the authors refer to "interactive stratospheric aerosol-climate models" (e.g. page 1, lines 20-21), and a potential variant of that 2nd suggested alternative could be to change the words "aerosol microphysics" for "aerosol-climate".

*We have adopted "An interactive stratospheric aerosol model intercomparison of solar geoengineering by stratospheric injection of $SO_2$ or accumulation-mode sulfuric acid aerosols" as the title. In lines 20-21 of the abstract we replace" aerosol-climate models" with "aerosol microphysics models".*

2) Abstract: Page 1, lines 12-14 -- this 1st line of the Abstract still seems poorly worded to me.
I don't understand what the authors are trying to communicate here. The wording currently states:

"Studies of stratospheric solar geoengineering have tended to focus on sulfuric acid aerosols, and almost all such climate model experiments assume that SO2 is injected to increase the sulfuric acid aerosol burden of the stratosphere."

I think this sentence should be replaced with a more meaningful comment on previous work, in relation to the difference between imposing a particular particle size, and simulating the size distribution interactively. This paper is the 1st ever inter-comparison of interactive stratospheric aerosol simulations of the geoengineering-enhanced stratospheric aerosol layer, and it is this functionality that I'd suggest this functionality the first sentence of the Abstract focuses on communicating.

A specific suggestion would be to replace that sentence with:

"Previous model comparisons of stratospheric solar geoengineering have mostly tended to focus on climate model experiments assuming a particular prescribed particle size for geoengineered sulfate aerosol particles."

The 1st part of the 2nd sentence of the Abstract should also be improved -- as it is similarly poorly worded at present -- my specific recommendation is to re-word that 1st part to instead begin "By contrast,
a key finding from interactive modeling studies is that the radiative forcing of a geoengineered stratospheric aerosol layer would increase sub-linearly....."

*The review misunderstands the meaning and intent of the first sentence of the abstract. We state that most studies of stratospheric solar geoengineering have focused on sulfuric acid aerosols (as opposed to solid particle injections) and that most of these studies (other than those that simply modify the solar constant or impose a stratospheric size distribution) assume injection of $SO_2$. The wording was not clear. We have made the wording in the first 3 sentences of the abstract much more precise:*

*"Studies of stratospheric solar geoengineering have tended to focus on modification of the sulfuric acid aerosol layer, and almost all climate model experiments that mechanistically increase the sulfuric acid aerosol burden assume injection of $SO_2$. A key finding from these model studies is that the radiative forcing would increase sub-linearly with increasing $SO_2$ injection because most of the added sulfur increases the mass of existing particles, resulting in shorter aerosol residence times and aerosols that are above the optimal size for scattering. Injection of $SO_3$ or $H_2SO_4$ from an aircraft in stratospheric flight is expected to produce particles predominantly in the accumulation-mode size range following microphysical processing within an expanding plume, and such injection may result in a smaller average stratospheric particle size, allowing a given injection of sulfur to produce more radiative forcing.."*

3) Abstract: Page 1, lines 28-30 -- The "We use" wording here is colloqial and needs to be changed, with also the "injection patterns" and "belt" somehow (to me at least) not sufficiently communicating the scientific issue being explored. The experiment that injects at 30N and 30S is presumably a specific "deployment scenario", aimed to force only the mid-latitudes, whereas the constant emissions from 30S-30N
is a more theoretical scenario, perhaps idealsed to achieving the longest residence time for the emitted particles. Suggest to re-word:

"We use two different injection patterns"

instead to

"The models carried out two different "geoengineering-enhancement scenarios" or similar more scientific terminology than "patterns".

I think there is an error here also where you state "injecting in a belt along the equator between 30S and 30N" --- perhaps it's simply a case of deleting "along the equator", but I'd recommend also changing "in a belt" to "uniformly" or "with a uniform emission rate" or similar.

*The wording here did need improvement. We replace "injection pattern" with "geographical distributions of injection mass", and add "idealized" in front of "geographical distribution". We also change "belt" to "region". And further clarify the 30S-30N injection to be "uniformly in the regiom between 30S and 30N" and motivated to "maximize aerosol residence time". The new working is:*

*"The model studies were carried out with two different idealized geographical distributions of injection mass representing deployment scenarios with different objectives, one designed to force mainly the midlatitudes by injecting into two grid points at 30° N and 30° S and the other designed to maximize aerosol residence time by injecting uniformly in the region between 30° S and 30° N."*

4) Abstract: Page 1, lines 30-32 -- This last sentence of the Abstract also needs to be improved, as it's not clear what is meant by "opposite impacts" -- and the term "radiative efficacy" seems somehow ill-defined (or not yet introduced). Perhaps a simple re-wording of "opposite impacts on radiative efficacy" to "strongly differing radiative forcing efficacy" -- or just "strongly differing radiative forcing".

*We agree that this sentence was not clear and that radiative efficacy is not yet defined in this context. Therefore we change this sentence to make our point about concentrated (2point) vs dispersed (regional) injection with reference to size distributions instead: "Analysis of aerosol size distributions in the perturbed stratosphere of the models shows that particle sizes evolve differently in response to concentrated versus dispersed injections depending on the form of the injected sulfur ($SO_2$ gas or AM-$H_2SO_4$ particulate) and suggests that prior model results for concentrated injection of $SO_2$ may be strongly dependent on model resolution" This concept is explained in Table 3 and we feel it should be highlighted in the abstract.*

5) Introduction: Page 3, lines 27-30 -- The current wording "Detailed modelling of this complex process for a full range of stratospheric physical, chemical and microphysical conditions awaits further studies" should be improved to better communicate the scientific issue here (rather than the technical aspects of the processes involved). I mean the question of how the particle size progresses, over a timescale of months to years, from the initial size at the plume-scale, to the global-scale variations in particle size in the "dispersed state" of the geoengineering-enhanced stratospheric aerosol layer.

Suggest instead "A priority for future modeling studies could potentially be to establish how the initial "engineered particle size" at the plume-scale progresses to the global-scale variations in particle size in the "dispersed state" of the geoengineering-enhanced stratospheric aerosol layer."

*This sentence refers to plume evolution, not large-scale evolution. We have revised it to now read: "Detailed modelling of potential plume-scale evolution under a full range of stratospheric physical, chemical, and microphysical conditions awaits further studies."*

The follow-on sentence should also be re-worded. The current text refers to "the GeoMIP models" but, further to my general comments above, I think the authors mean the interactive stratospheric aerosol
models -- i.e. "the ISA-MIP models". Also change "can be used to simulate" to "have the functionality to
explore how the stratospheric aerosol layer responds with the global dispersion of the geoengineering particles." Or something like this.

*We have added the reference to Timmreck et al., 2018 and modify this sentence as suggested: "For the temporal and spatial scale beyond plume models, global GCMs such as those participating in the Interactive Stratospheric Aerosol Model Intercomparison Project (ISA-MIP, Timmreck et al., 2018) have the functionality to explore how the stratospheric aerosol layer responds with global dispersion of the geoengineering injections." In the next paragraph, after naming the three models participating in this study, we add: "These three models are participants in both the GeoMIP and ISA-MIP model intercomparisons."*

6) Introduction: Page 3, lines 30-33 -- The 1st of the 2 setnences here refers to ozone, temperature and circulation -- but the impacts on ozone depends strongly on the stratospheric chemistry scheme, which is outside the scope of this article. I suggest to narrow the scope of this sentence to instead to focus on the aerosol changes. A specific suggestion is to change:

"These GCMs can effectively simulate changes in global aerosol
burden, radiative forcing, ozone, and stratospheric temperature and circulation

instead to

"Those ISA-MIP models with microphysical aerosol schemes can also address the key issue of how the particle size distribution progressees, this being a key determinant of subsequent global aerosol burden and radiative forcing."

As in my general/overarching comments above, the 2nd sentence here also needs changing, with this issue of an "input size distribution" needing to be better explained.
The wording says:

"As input, they would take the particle size distribution from aircraft plume model studies
but could represent any hypothetical input of particles".

I think this reference to an "input size distribution" could be mis-interpreted by some readers unfamiliar with the types of model involved. The value of the interactive stratospheric aerosol models is more than simply that they can represent an input size from a plume-scale model. It's this issue of how the emitted particles subsequently transform in a "globally dispersed state", as they become part of a geoengineering-enhanced stratospheric aerosol layer.

*We adopt much of the suggested rewording in this paragraph. "Those models with microphysical aerosol schemes can also address the key issue of how the particle size distribution evolves, this being a key determinant of subsequent global aerosol burden and radiative forcing. As input, the global models would take a mass flux of particles with the size distributions generated by aircraft plume model studies, or any hypothetical source of particles at a specified rate. The input size distribution is simplified here by using a lognormal distribution with a constant mode radius*

*and mode width for all injection grid points and times."*

*By clarifying that the geoengineering particle injections involve both a size distribution and a mass flux rate, we hope to avoid misinterpretation that a size distribution is imposed on the global stratosphere.*

With the re-wordings in points 1 to 5, it may be that this is then sufficiently explained, with then this final sentence explaining of the potential for future work to involve a combination of plume-scale models and the global interactive stratospheric aerosol GCMs.

A specific suggestion for re-wording could be:

"There is the potential for future interactive stratospheric aerosol model experiemnts to link directly with plume-scale model experiments, and seek to realistically represent potential alternative deployment scenarios".

*We have added this statement to the last paragraph of the summary and discussion section which mentioned plume-in-grid and adaptive mesh as methods of combining plume and global scale modeling. "Improved understanding of the effectiveness of stratospheric sulfur injection and the role of plume-scale formation of accumulation mode particles may require use of modelling methods such as plume-in-grid or adaptive mesh to better capture the multi-scale problem from injection plume to the global circulation. Such methods may allow future interactive stratospheric aerosol model experiments to link directly with plume-scale model experiments, and seek to realistically represent potential alternative deployment scenarios."*

References
* * *
Timmreck, C., Mann, G.W., Aquila, V., Hommel. R., Lee, L. A., Schmidt, A. et al. (2018):
"The Interactive Stratospheric Aerosol Model Intercomparison
Project (ISA-MIP): motivation and experimental design"
Geosci. Model Dev., 11, 2581–2608.
https://doi.org/10.5194/gmd-11-2581-2018

*Reference added.*